# Scaling Image and Video Generation via Test-Time Evolutionary Search

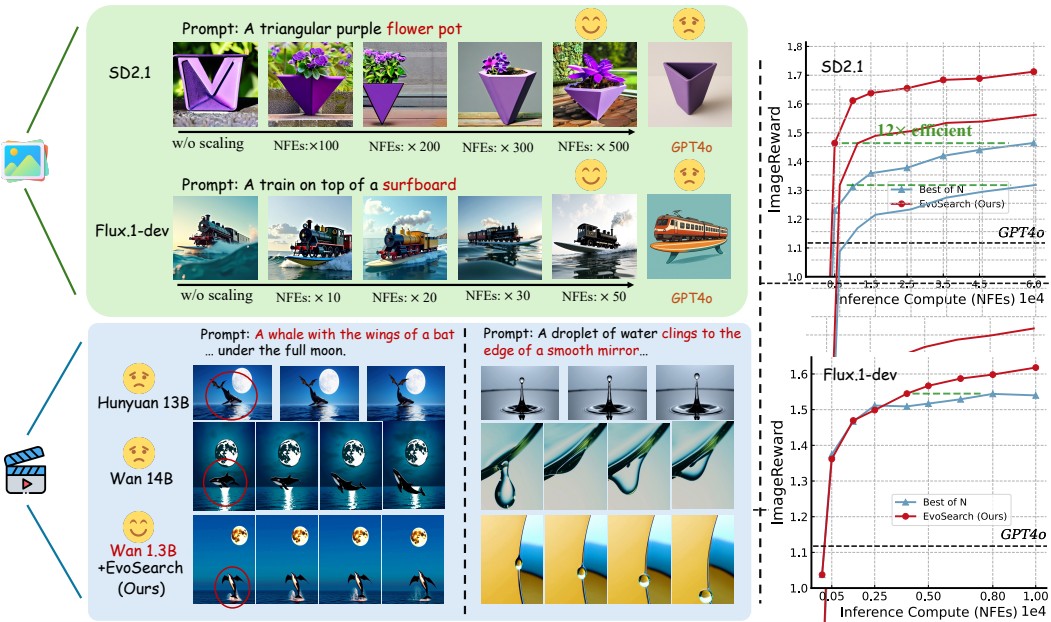

Figure 1: We propose **Evol**utionary **Search** (EvoSearch), a novel, generalist, and compute-optimal test-time scaling framework applicable to both image and video generation tasks. EvoSearch significantly enhances sample quality through strategic computation allocation during inference, **enabling Stable Diffusion 2.1 to be comparable to GPT4o, and Wan 1.3B to outperform Wan 14B model and Hunyuan 13B model with $10\times$ fewer parameters**.

## Abstract

As the marginal cost of scaling computation (data and parameters) during model pre-training continues to increase substantially, test-time scaling (TTS) has emerged as a promising direction for improving generative model performance by allocating additional computation at inference time. While TTS has demonstrated significant success across multiple language tasks, there remains a notable gap in understanding the test-time scaling behaviors of image and video generative models (diffusion-based or flow-based models). Although recent works have initiated exploration into inference-time strategies for vision tasks, these approaches face critical limitations: being constrained to task-specific domains, exhibiting poor scalability, or falling into reward over-optimization that sacrifices sample diversity. In this paper, we propose **Evol**utionary **Search** (EvoSearch), a novel, generalist, and efficient TTS method that effectively enhances the scalability of both image and video generation across diffusion and flow models, without requiring additional training or model expansion. EvoSearch reformulates test-time scaling for diffusion and flow models as an evolutionary search problem, leveraging principles from biological evolution to efficiently explore and refine the denoising trajectory. By incorporating carefully designed selection and mutation mechanisms tailored to the stochastic differential equation denoising process, EvoSearch iteratively generates higher-quality offspring while preserving population diversity. Through extensive evaluation across both diffusion and flow architectures for image and video generation tasks, we demonstrate that our method consistently outperforms existing approaches, achieves higher diversity, and shows strong generalizability to unseen evaluation metrics.

# 1 INTRODUCTION

Generative models have witnessed remarkable progress across various fields, including language (Achiam et al., 2023; Jaech et al., 2024; Guo et al., 2025), image (Esser et al., 2024; Labs, 2024), and video generation (Brooks et al., 2024; Kong et al., 2024; Wang et al., 2025), demonstrating powerful capabilities to capture complex data distributions. The central driver of this success is their ability to scale up during training by increasing data volumes, computational resources, and model sizes. This scaling behavior during the training process is commonly described as Scaling Laws (Hoffmann et al., 2022; Kaplan et al., 2020). Despite these advancements, further scaling at training time is increasingly reaching its limits due to the rapid depletion of available internet data and increasing computational costs. Post-training alignment (Tie et al., 2025) has been proven to be effective in addressing this challenge. For diffusion and flow models, these approaches typically include parameter tuning via reinforcement learning (Black et al., 2023; Fan et al., 2023) or direct reward gradient backpropagation (Clark et al., 2024; Prabhudesai et al., 2024). However, they suffer from reward over-optimization due to their mode-seeking behavior, high computational costs, and requirement of direct model weight access. Alternative methods (Ahn et al., 2024; Zhou et al., 2024) propose directly optimizing initial noise, as some lead to better generations than others, but demand specialized training and struggle with cross-model generalization.

Recent advances in large language models (LLMs) have expanded to test-time scaling (TTS) (Brown et al., 2024; Wu et al., 2025), showing promising results to complement traditional training-time scaling law. TTS (Zhang et al., 2025) allocates additional computation budget during inference, offering a novel paradigm for improving generation quality without additional training. However, diffusion and flow models present unique challenges for test-time scaling, since they must navigate the complex, high-dimensional state space along the denoising trajectory, where existing methods in LLMs struggle to transfer effectively. Current approaches of test-time scaling for diffusion and flow models include (i) best-of-N sampling (Ma et al., 2025; Liu et al., 2025a), which, despite its simplicity, suffers from severe search inefficiency in high-dimensional noise spaces; and (ii) particle sampling (Kim et al., 2025b; Singhal et al., 2025a), which, while enabling search across the entire denoising trajectory, compromises both exploration capability and generation diversity due to its reliance on initial candidate pools. These simple heuristic designs lack fundamental adaptability to the complex generation pathways, leading to sample diversity collapse and inefficient computation.

In this paper, we aim to address the above critical challenges and develop a general and efficient test-time scaling method that is versatile for both image and video generation across diffusion and flow models without parameter tuning or gradient backpropagation. To enable test-time scaling of flow models, we transform their deterministic sampling process (ODE) into a stochastic process (SDE), thereby broadening the generation space, which paves the way for a unified framework for inference-time optimization. Through systematic analysis of latent spaces along the denoising trajectory, including both starting Gaussian noises and intermediate states, we find that neighboring states in the latent space exhibit similar generation qualities, suggesting that high-quality samples are not solely isolated. Based on this insight, we propose **Evo**lutionary **Search** (EvoSearch), a novel test-time scaling method inspired by biological evolution. EvoSearch reframes test-time scaling of image and video generation as an evolutionary search problem, incorporating selection and mutation mechanisms specifically designed for the denoising process in both diffusion and flow models. At each generation, EvoSearch first selects high-reward parents while preserving population diversity, and then generates new offspring through our designed denoising-aware mutation mechanisms to explore new states, enabling iterative improvement in sample quality. The key insight of EvoSearch is to actively explore high-reward particles through evolutionary mechanisms, overcoming the limitations of previous search methods that are confined to a fixed candidate space. To optimize computational efficiency, we dynamically search along the denoising trajectory, progressing from Gaussian noises to states at larger denoising steps, thereby continuously reducing computational costs as we approach the terminal states. Through extensive experiments on both text-conditioned image generation and video generation tasks, we find that EvoSearch achieves substantial improvements in sample quality and human-preference alignment as test-time compute increases.

We summarize our key contributions as follows: (i) We propose EvoSearch, a novel, generalist, and efficient TTS framework which enhances generation quality by allocating more compute during inference, unifying optimization for both diffusion and flow generative models. (ii) Based on our observations of latent space structure, we design specialized selection and mutation mechanisms tailored to the denoising process, effectively enhancing exploration while maintaining diversity. (iii) Extensive experiments show that EvoSearch effectively improves generative model performance by scaling up inference-time compute, outperforming competitive baselines across both image and video

generation tasks. Notably, EvoSearch enables SD2.1 (Dhariwal & Nichol, 2021) to be comparable to GPT4o, and allows the Wan 1.3B model (Wang et al., 2025) to achieve competitive performance with the $10\times$ larger Wan 14B model. Our project is available at evosearch.github.io.

## 2 PRELIMINARY

**Diffusion Models and ODE-to-SDE Transformation of Flow Models.** Both diffusion models and flow models map the source distribution, often a standard Gaussian distribution, to a true data distribution $p_0$. A forward diffusion process progressively perturbing data to noise, defined as $\boldsymbol{x}_t = \alpha_t \boldsymbol{x}_0 + \sigma_t \varepsilon$, where $\varepsilon \in \mathcal{N}(0, I)$ is the added noise at timestep $t \in [0, T]$, and $(\alpha_t, \sigma_t)$ denote the noise schedule. To restore from diffused data, diffusion models naturally utilize an SDE-based sampler during inference (Song et al., 2020b;a), which introduces stochasticity at each denoising step as follows: $\boldsymbol{x}_{t-1} = \sqrt{\alpha_{t-1}} \left( (\boldsymbol{x}_t - \sqrt{1 - \alpha_t} \varepsilon_\theta(\boldsymbol{x}_t, t)) / \sqrt{\alpha_t} \right) + \sqrt{1 - \alpha_{t-1} - \sigma_t^2} \varepsilon_\theta(\boldsymbol{x}_t, t) + \sigma_t \varepsilon_t$. In contrast, flow models learn the velocity $u_t \in \mathbb{R}^d$, which enables sampling of $\boldsymbol{x}_0$ by solving the flow ODE (Song et al., 2020b) backward from $t = T$ to $t = 0$: $\boldsymbol{x}_{t-1} = \boldsymbol{x}_t + u_t(\boldsymbol{x}_t) dt$, leading all $\boldsymbol{x}_{t-1}$ drawn from $\boldsymbol{x}_t$ identical. This restricts the applicability of test-time scaling search methods like particle sampling and our proposed EvoSearch in flow models (Kim et al., 2025a), since the sampling process lacks stochasticity beyond initial noise. To address this limitation, we transform the deterministic Flow-ODE into an equivalent SDE process. Following previous works (Albergo et al., 2023; Ma et al., 2024; Patel et al., 2024; Kim et al., 2025a; Singh & Fischer, 2024), we rewrite the ODE sampling process by $d\boldsymbol{x}_t = \left( u_t(\boldsymbol{x}_t) - \frac{\sigma_t^2}{2} \nabla \log p_t(\boldsymbol{x}_t) \right) dt + \sigma_t d\boldsymbol{w}$, where the score $\log p_t(\boldsymbol{x}_t)$ can be computed by velocity $u_t$ (see Eq. (13) in (Singh & Fischer, 2024)), and $d\boldsymbol{w}$ injects stochasticity at each sampling step.

**Evolutionary Algorithms.** Evolutionary algorithms (EAs) (Koza, 1992; Bäck, 1996) are biologically inspired, gradient-free methods that found effective in optimization (Goldberg, 1989; Grefenstette, 1993; Vikhar, 2016), algorithm search (Co-Reyes et al., 2021; Real et al., 2020), and neural architecture search (Real et al., 2019; Yang et al., 2020; So et al., 2019). The key idea of EAs is mimicking the process of natural evolution (Ao, 2005), by maintaining a population of solutions that evolve over generations. EAs involve initializing random solutions, evaluating fitness, selecting parents, and applying genetic operators (crossover and mutation) to create offspring that constitute the next generation. Due to the diversity within populations and the mutation operations, EAs excel at global optimization and solving multimodal problems compared to traditional local search methods.

## 3 RELATED WORK

**Alignment for Diffusion and Flow Models.** Aligning pre-trained diffusion and flow generative models can be achieved by guidance (Dhariwal & Nichol, 2021; Song et al., 2020b) or fine-tuning (Lee et al., 2023; Fan & Lee, 2023), which aim to enhance sample quality by steering outputs towards a desired target distribution. Guidance methods (Ho et al., 2022; Song et al., 2023a; Chung et al., 2023; Bansal et al., 2023; Song et al., 2023b; Guo et al., 2024b) rely on predicting clean samples from noisy data and differentiable reward functions to calculate guidance. Typical fine-tuning methods involve supervised fine-tuning (Lee et al., 2023; ?; Fan & Lee, 2023; Wu et al., 2023c), RL fine-tuning (Black et al., 2023; Fan et al., 2023; Liu et al., 2025b; Miao et al., 2024), DPO-based policy optimization (Wallace et al., 2024; Yang et al., 2024; Liang et al., 2024; Liu et al., 2024; Zhang et al., 2024), direct reward backpropagation (Clark et al., 2024; Xu et al., 2023; Prabhudesai et al., 2024), stochastic optimization (Domingo-Enrich et al., 2024; Yeh et al., 2024), and noise optimization (Ahn et al., 2024; Zhou et al., 2024; Tang et al., 2024; Guo et al., 2024a; Eyring et al., 2024). Previous works (Miao et al., 2024) calculate reward signals on fully denoised samples for ensuring trustworthy feedback. However, these methods require additional dataset curation and parameter tuning, and can distort alignment or reduce sample diversity due to their mode-seeking behavior and reward over-optimization. In contrast, our proposed EvoSearch method offers significant advantages through its universal applicability across any reward function and model architecture (including flow-based, diffusion-based, image, and video models) without requiring additional training. Moreover, EvoSearch complements existing fine-tuning methods, as it can be applied to any fine-tuned model to further enhance reward alignment. While a related work (Domingo-Enrich et al., 2024) demonstrates that SDE dynamics are critical for fine-tuning diffusion models, our work is distinctive since we focus on the test-time inference phase instead of the training phase with parameter updates.

**Test-Time Scaling in Vision.** Several test-time scaling (TTS) methods have been proposed to extend the performance boundaries of image and video generative models. These methods fundamentally

operate as search, with reward models providing judgments and algorithms selecting better candidates. Best-of-N generates N batches of samples and selects the one with the highest reward, which has been validated effective for both image and video generation (Ma et al., 2025; Liu et al., 2025a). More advanced search method for diffusion models is particle sampling (Singhal et al., 2025a; Li et al., 2024; 2025; Singh et al., 2025; Kim et al., 2025b), which resamples particles over the full denoising trajectory based on their importance weights, demonstrating superior results than naive BoN. Video-T1 (Liu et al., 2025a) and other recent works (Yang et al., 2025; Xie et al., 2025; Oshima et al., 2025; Liu et al., 2025a) propose leveraging beam search (Snell et al., 2024) for scaling video generation. However, in the context of diffusion and flow models, we remark that beam search represents a specialized case of particle sampling with a predetermined beam size, as both methodologies iteratively propagate high-reward samples while discarding lower-reward ones in practice. Furthermore, Video-T1 is constrained to autoregressive video models, limiting its applicability to more advanced diffusion and flow generative models. All existing search methods rely heavily on passive filtering, failing to explore new particles actively, while our proposed method, EvoSearch, leverages the idea of natural selection and evolution, enabling the generation of new, higher-quality offspring iteratively. EvoSearch is also a generalist framework with superior scalability and extensive applicability across both diffusion and flow models for image and video generation, contrary to previous methods that are constrained to specific models or tasks.

## 4 PROPOSED METHOD

### 4.1 PROBLEM FORMULATION

In this work, we investigate how to efficiently harness additional test-time compute to enhance the sample quality of image and video generative models. Given a pre-trained flow-based or diffusion-based model and a reward function, our objective is to generate samples from the following target distribution (Uehara et al., 2024a; Li et al., 2024; Wu et al., 2023a; Uehara et al., 2024b):

$$p^{\text{tar}} = \frac{1}{\mathcal{Z}} p_0^{\text{pre}}(\boldsymbol{x}_0) \exp\big(\frac{r(\boldsymbol{x}_0)}{\alpha}\big),\tag{1}$$

where $\mathcal{Z}$ denotes a normalization constant (Rafailov et al., 2023; Uehara et al., 2024a) and $p_0^{\text{pre}}$ is the pre-trained distribution. Notably, directly sampling from the target distribution is infeasible: the normalization factor $\mathcal{Z}$ requires integrating over the entire sample space, making it computationally intractable for high-dimensional spaces in diffusion and flow models.

### 4.2 LIMITATIONS OF EXISTING APPROACHES

Test-time approaches to sampling from the target distribution $p^{\text{tar}}$ in Eq. equation 1 employ importance sampling (Owen & and, 2000), which generates $k$ particles $\boldsymbol{x}_0^i \sim p_0^{\text{pre}}(\boldsymbol{x}_0)$ and then resamples the particles based on the scores $\exp(r(\boldsymbol{x}_0)/\alpha)$. A straightforward implementation of this concept is best-of-N sampling, which simply generates multiple samples and selects the one with the highest reward. A more sophisticated approach, called particle sampling (Singhal et al., 2025b; Kim et al., 2025b), searches across the entire denoising path $\tau = \{\boldsymbol{x}_T, \cdots, \boldsymbol{x}_k, \cdots, \boldsymbol{x}_0\}$, guiding samples toward trajectories that yield higher rewards. However, both of these methods suffer from fundamental limitations in their efficiency and exploration capabilities. Best-of-N only resamples at the final step ($t = 0$), taking the entire distribution $p_0^{\text{pre}}(\boldsymbol{x}_0) = \int \prod_t \{p_t^{\text{pre}}(\boldsymbol{x}_{t-1}|\boldsymbol{x}_t)\} d\boldsymbol{x}_{1:T}$ as its proposal distribution. This passive filtering approach is computationally wasteful, as it expends a large amount of computation generating complete trajectories for samples that ultimately yield low rewards. In contrast, particle sampling can search and resample at each intermediate step along the denoising path, using $p_t^{\text{pre}}(\boldsymbol{x}_{t-1}|\boldsymbol{x}_t)$ as its proposal distribution at each step $t$. However, it is still constrained by the fixed initial candidate pool, struggling to actively explore and generate novel states beyond those proposed by $p_0^{\text{pre}}$ during the search process. This limitation becomes increasingly restrictive as the search progresses, which leads to restricted performance due to limited exploration and reduced diversity.

To better understand these inherent limitations more concretely, we visualize the behavior of different approaches in Fig. 2. As shown, re-training methods, including RL (DDPO (Black et al., 2023)) and reward backpropagation (Clark et al., 2024), struggle to generalize to the unseen target distribution, largely due to their heavy reliance on pre-trained models and mode-seeking behavior. While test-time search methods (best-of-N and particle sampling) achieve higher rewards than re-training methods, they still fail to capture all modes of the multimodal target distribution, converging to limited regions of the solution space. These findings highlight the need for a novel test-time scaling framework capable of effectively balancing between exploitation and exploration while maintaining computational

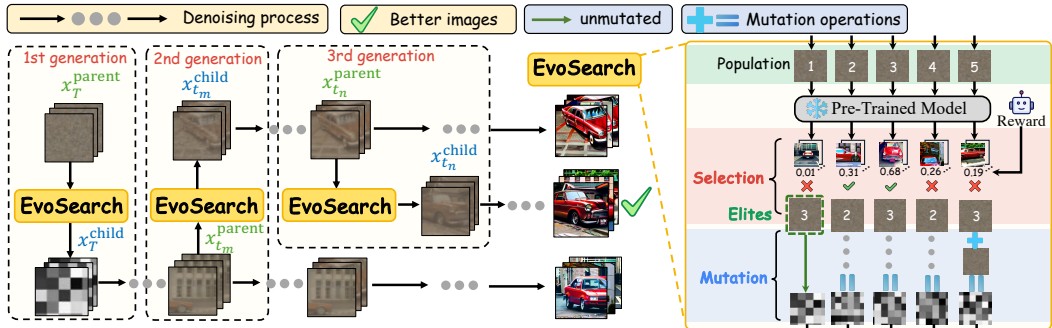

Figure 2: Visualization of a test-time alignment experiment. We train a diffusion model with 3-layer MLP on Gaussian mixtures (pre-trained distribution), with the goal to capture multimodal unseen target distribution, where reward $r(X, Y) = -|X^2 + Y^2 - 4|$. EvoSearch achieves superior performance, capturing all the modes with the highest reward (-0.74).

Figure 3: Overview of our method. EvoSearch progressively moves forward along the denoising trajectory to refine and explore new states. At each evolution step defined by $\mathcal{T}$, our proposed *EvoSearch* process generates novel, high-quality offspring $x_T^{\text{child}}$ based on the parent population $x_t^{\text{parent}}$. The population size across each generation is defined by schedule $\mathcal{K}$. *EvoSearch* contains evaluation, selection, and mutation operations to ensure the effectiveness of the evolutionary process. The generation quality consistently improves with the progression of the evolutionary search.

efficiency for scaling up. In the following sections, we introduce how our EvoSearch method overcomes these fundamental limitations, which achieves the highest reward with comprehensive mode coverage as shown in Fig. 2.

## 4.3 EVOLUTIONARY SEARCH

We propose **Evo**lutionary **Search** (EvoSearch), a novel evolutionary framework that reformulates the sampling from the target distribution $p^{\text{tar}}$ in Eq. 1 at test time as an active evolutionary optimization problem rather than passive filtering. EvoSearch introduces a unified way for achieving efficient and effective test-time scaling across both diffusion and flow models for image and video generation tasks. The overview of our method is provided in Fig. 3. Our algorithm is summarized in Alg 1&2. EvoSearch introduces a novel perspective that reinterprets the denoising trajectory as an evolutionary path, where both the initial noise $x_T$ and the intermediate state $x_t$ can be evolved towards higher-quality generation, actively expanding the exploration space beyond the constraints of the pre-trained model's distribution. Below, we introduce the core components of EvoSearch.

**Evolution Schedule.** For a typical sampling process in diffusion and flow models, the change between $x_{t-1}$ and $x_t$ is not substantial. Therefore, performing EvoSearch at every sampling step would be computationally wasteful. To address this efficiency problem, EvoSearch defines an evolution schedule $\mathcal{T} = \{T, \cdots, t_m, \cdots, t_n\}$ that specifies the timesteps at which EvoSearch should be conducted. Concretely, EvoSearch first thoroughly optimizes the starting noise $x_T$ to identify high-reward regions in the Gaussian noise space, establishing a strong initialization for the subsequent denoising process. After a high-quality $x_T$ is obtained, EvoSearch progressively applies our proposed evolutionary operations to intermediate states $x_{t_i}$ at predetermined timesteps $t_i \in \mathcal{T}$. This cascading way enables each subsequent generation beginning directly from the cached intermediate state $x_{t_i}$ obtained from the previous generation, instead of repeatedly denoising from $x_T$, eliminating the redundant denoising computations from $x_T \to x_{t_i}$. In practice, we implement this evolution schedule using uniform intervals between timesteps, which significantly reduces computational overhead.

**Population Initialization.** Following the evolution schedule $\mathcal{T}$, we introduce a corresponding population size schedule $\mathcal{K} = \{k_T, \cdots, k_m, \cdots, k_n\}$, where each $k_i$ specifies the population size for the generation at timestep $t_i$. This adaptive approach enables flexible trade-offs between computational cost and exploration of the state space (Appendix B.1 for further analysis on ablation of $\mathcal{K}$ and $\mathcal{T}$). The initial generation of EvoSearch begins with $k_T$ randomly sampled Gaussian noises $\{x_T^i\}_{i=1}^{k_T}$ at timestep $t = T$, which serve as the first-generation parents for the subsequent evolutionary process.

**Fitness Evaluation.** To guide the evolutionary process, EvoSearch evaluates the quality of each parent using an off-the-shelf reward model at each evolution timestep $t_i$:

$$R(\boldsymbol{x}_{t_i}) = \mathbb{E}_{\boldsymbol{x}_0 \sim p_0(\boldsymbol{x}_0 | \boldsymbol{x}_{t_i})} \left[ r(\boldsymbol{x}_0) | \boldsymbol{x}_{t_i} \right], \tag{2}$$

where the reward model $r$ can correspond to various objectives, including human preference scores (Xu et al., 2023; Wu et al., 2023b; Hessel et al., 2021) and vision-language models (Liu et al., 2025a; He et al., 2024). Note that previous methods typically rely on either lookahead estimators (Oshima et al., 2025; Li et al., 2025) or Tweedie's formula (Efron, 2011; Kim & Ye, 2021) to predict $\boldsymbol{x}_0$ from noisy data for reward calculation in Eq. equation 2, which can induce significant prediction inaccuracies and approximation errors. In contrast, we evaluate the reward directly on fully denoised $\boldsymbol{x}_0$ (e.g., clean image or video), thereby obtaining high-fidelity reward signals. Considering computing the exact expectation is computationally prohibitive, as it would require generating multiple full trajectories for every candidate at each intermediate denoising step. Therefore, in our implementation, we use a single-sample Monte Carlo approximation, estimating the expectation based on a single sample $x_0$. Although this is an estimate of the true expectation, the evolutionary population dynamics allow EvoSearch to robustly optimize the objective despite the variance in individual estimates.

**Selection.** To propagate high-quality candidates across generations while maintaining population diversity, EvoSearch employs tournament selection (Goldberg & Deb, 1991) to sample parents from the population of size $k_i$ through cycles. Specifically, each cycle picks a tournament of $b < k_i$ candidates at random and selects the best candidate in the tournament as a parent.

**Mutation.** Recent works (Zhou et al., 2024; Ahn et al., 2024) have shown that different initial noises yield varying generation quality. Intuitively, this property extends naturally to intermediate denoising states. While this phenomenon serves as a basis for making best-of-N and particle sampling useful, it raises a more fundamental question: *do these noises and intermediate states possess other exploitable patterns or structural regularities that can be leveraged to enhance inference-time generation quality?*

To investigate this critical question, we visualize the latent states at different denoising steps using t-SNE (Van der Maaten & Hinton, 2008). Our findings, as shown in Fig. 4, reveal that neighboring states in the latent space exhibit similar generation qualities, suggesting that high-quality samples are not solely isolated. Building upon this discovery, we develop a specialized mutation strategy that leverages this exploitable structure in the reward landscape of diffusion and flow models. Specifically, we preserve $m$ elite parents (those with top fitness scores) at each generation to ensure convergence, where $m \ll k_i$. For the remaining $k_i - m$ parents, we mutate them to explore the neighborhoods around selected parents to discover higher-quality samples. This approach avoids premature convergence to a narrow region of the denoising state space, facilitating effective exploration of novel regions while maintaining population diversity. To align with the characteristics of the underlying SDE sampling process, we develop different mutation operations for initial noises and intermediate denoising states.

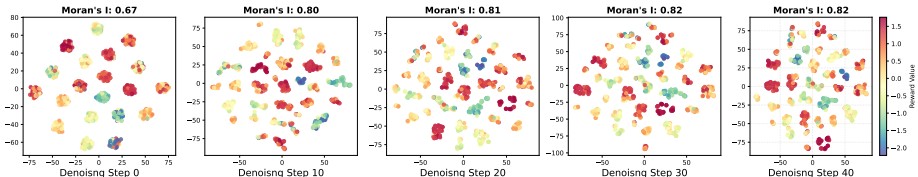

Figure 4: t-SNE Visualization of latent $\boldsymbol{x}_t$ from SD2.1 model at different steps, colored by their corresponding ImageReward scores. At denoising step 0, $\boldsymbol{x}_t$ is Gaussian noise. High-reward states exhibit significant spatial correlations, as measured by Moran's I (Moran, 1950).

● **Initial noise mutation.** For the initial noise $\boldsymbol{x}_T$, which is sampled from a Gaussian distribution, the corresponding mutation operation is designed to preserve the Gaussian nature of the noise based on

$$\boldsymbol{x}_T^{\text{child}} = \sqrt{1 - \beta^2} \boldsymbol{x}_T^{\text{parent}} + \beta \varepsilon_T, \quad \varepsilon_T \sim \mathcal{N}(0, I), \tag{3}$$

where $\beta$ controls the added stochasticity to the parents. The first term ensures that the mutated children preserve the high-reward region density, while the second term encourages exploration.

● **Intermediate denoising state mutation.** For intermediate states $\boldsymbol{x}_t$, the mutation operation defined in Eq. 3 is not applicable since $\boldsymbol{x}_t$ is no longer Gaussian due to the denoising process. To synthesize meaningful variations while preserving the intrinsic structure of the latent state $\boldsymbol{x}_t$, we propose an alternative mutation operator inspired by the reverse-time SDE:

$$\boldsymbol{x}_{t-1}^{\text{child}} = \boldsymbol{x}_{t-1}^{\text{parent}} + \sigma_t \varepsilon_t, \quad \varepsilon_t \sim \mathcal{N}(0, I), \tag{4}$$

where $\sigma_t$ is the diffusion coefficient defined in reverse-time SDE, controlling the level of injected stochasticity. This mutation operation effectively generates novel $x_{t-1}$, enabling exploration of an expanded state space while preserving the inherent distribution established during the denoising process. The theoretical validation of the proposed mutation strategies is provided in the Appendix A.2. In the next generation of EvoSearch, we sample $x_0 \sim p_0(x_0|x_t^{\text{child}})$ based on the new offspring $x_t^{\text{child}}$, and repeat the above evolutionary search process, including evaluation, selection, and mutation. We highlight that EvoSearch provides a unified framework that encompasses both best-of-N and particle sampling as special cases.

## 5 EXPERIMENTS

In this section, we evaluate the efficacy of EvoSearch through extensive experiments on large-scale text-conditioned generation tasks, encompassing both image and video domains.

### 5.1 EXPERIMENT SETUP

**Image Generation.** (i) **Tasks and Metrics.** We adopt DrawBench (Saharia et al., 2022) for evaluation, which consists of 200 prompts spanning 11 different categories. We utilize multiple metrics to evaluate generation quality, including ImageReward (Xu et al., 2023), HPSv2 (Wu et al., 2023b), Aesthetic score (Schuhmann et al., 2022), and ClipScore (Hessel et al., 2021). ImageReward and ClipScore are employed as guidance rewards during search. Please refer to evaluation details in Appendix A.3. (ii) **Models.** We employ two different text-to-image models to evaluate EvoSearch and baselines, which are Stable Diffusion 2.1 (Rombach et al., 2022) and Flux.1-dev (Labs, 2024), respectively. SD2.1 is a diffusion-based text-to-image model with 865M parameters, while Flux-dev is a rectified flow-based model with 12B parameters. For both models, we use 50 denoising steps with a guidance scale of 5.5, with other hyperparameters remaining as the default.

**Video Generation.** (i) **Tasks and Metrics.** We take the recently released VideoReward (Liu et al., 2025b) as the guidance reward to provide feedback during search. VideoReward, built on Qwen2-VL-2B (Wang et al., 2024), evaluates generated videos on multiple dimensions: visual quality, motion quality, and text alignment. To measure the generalization performance to unseen rewards, we utilize both automatic metrics and human assessment for comprehensive evaluation. For automatic evaluation, we employ multiple metrics from VBench (Huang et al., 2024) and VBench2 (Zheng et al., 2025), which encompass 625 distinct prompts distributed across six fundamental dimensions, including *dynamic*, *semantic*, *human fidelity*, *composition*, *physics*, and *aesthetic*. For human evaluation, we hire annotators to evaluate videos on 200 prompts sampled from VideoGen-Eval (Zeng et al., 2024). Evaluation details are in Appendix A.3.

(ii) **Models.** To evaluate the scalability and performance of baselines, we utilize two widely adopted video generative models: HunyuanVideo (Kong et al., 2024) and Wan (Wang et al., 2025). Given the computational intensity of video generation compared to image generation, we specifically use the 1.3B parameter variant of Wan for practical evaluation. Each video comprises 33 frames, with other hyperparameters following default configurations.

**Baselines.** As we evaluate the scalability of both diffusion and flow models across image and video generation tasks, we benchmark EvoSearch against two widely-used search methods that are applicable to our experimental settings: (i) **Best of N** samples multiple random noises at beginning, assign reward values to them via denoising and evaluation, and choose the candidate yielding the highest reward. (ii) **Particle Sampling** follows the implementation of FK-Steering (Singhal et al., 2025a), which maintains a set of candidates along the denoising process, called particles, and iteratively propagates high-reward samples while discarding lower-reward ones. Im-

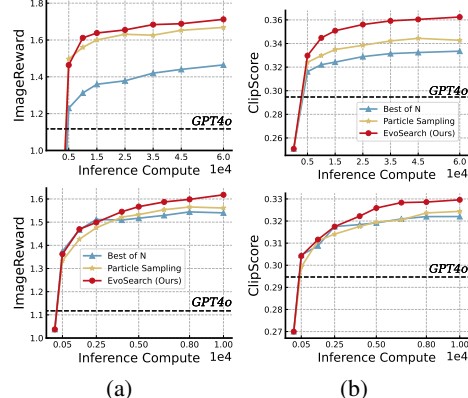

Figure 5: Scaling behavior of EvoSearch and baselines as inference-time computation increases on DrawBench. *Top*: SD2.1. *Bottom*: Flux.1-dev. (a) and (b) use ImageReward and ClipScore as guidance rewards, respectively.

plementation details of EvoSearch and baselines are provided in Appendix A.1. To ensure fair comparison, we employ the same random seeds to generate videos/images for each method.

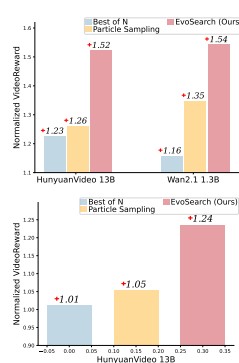

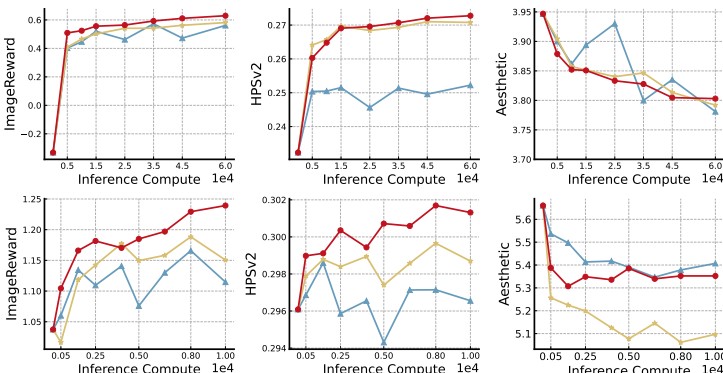

Figure 6: VideoRewards on VBench & VBench2.0 (*top*) and VideoGen-Eval (Zeng et al., 2024) (*bottom*).

Figure 7: EvoSearch can generalize to unseen metrics. *Top row*: DrawBench results on SD2.1. *Bottom row*: DrawBench results on Flux.1-dev.

## 5.2 RESULTS ANALYSIS

To evaluate EvoSearch's versatility and practical performance, we include image generation on diffusion model (SD2.1) and flow model (Flux.1-dev), video generation on flow models (HunyuanVideo and Wan) for comprehensive empirical analysis. In addition to comparing the number of function evaluations (NFEs), we provide a wall-clock time comparison in the Appendix B.2 to further demonstrate the computational efficiency of our method, wherein the advantages of EvoSearch become even more evident.

**Question 1.** *Can EvoSearch consistently yield performance improvement with scaled inference-time computation?*

As shown in Fig. 5, where we evaluate performance using both ImageReward and ClipScore, EvoSearch exhibits monotonic performance improvements with increasing inference-time computation. Notably, for the Flux.1-dev model (12B parameters), EvoSearch continues to demonstrate performance gains as NFEs increase, whereas baseline methods plateau after approximately $1e4$ NFEs. Qualitative results in Fig. 1 show that both SD2.1 and Flux.1-dev generate images with progressively improved prompt alignment as inference computation (i.e., NFEs) increases.

**Question 2.** *How does EvoSearch compare to baselines for scaling image and video generation at inference time?*

For image generation tasks, as evidenced in Fig. 5 and Fig. 7, EvoSearch demonstrates consistent superior performance over all baseline methods across varying computational budgets, for both diffusion-based SD2.1 and flow-based Flux.1-dev models. The results on other benchmarks like GenEval (Ghosh et al., 2023) and DPGBench (Hu et al., 2024) are provided in Appendix B.3. For video generation tasks where VideoReward serves as the guidance reward, EvoSearch continues to obtain the highest score across different generative models compared to the baselines. Quantitative results in Fig. 6 (top row) show that for the Wan 1.3B model, EvoSearch outperforms best-of-N and particle sampling by 32.8% and 14.1%, respectively. When applied to the larger HunyuanVideo 13B model, EvoSearch demonstrates improvements of 23.6% and 20.6% over best-of-N and particle sampling, respectively. Results on the prompts sample from Videogen-Eval (Zeng et al., 2024), as illustrated in Fig. 6 (bottom row), further corroborate these findings, with EvoSearch showing improvements of 22.8% and 18.1% compared to best-of-N and particle sampling, respectively. Qualitative assessment in Fig. 8 reveals that only EvoSearch successfully generates images with both background consistency and accurate text prompt alignment. In contrast, particle sampling fails to comprehend the complex text prompt, while best-of-N produces results of inferior visual quality. More qualitative results are provided in Appendix B.4. The superior performance of EvoSearch can be attributed to its active exploration and refinement within the denoising state space, whereas best-of-N and particle sampling are limited to a local candidate pool.

**Question 3.** *How does EvoSearch generalize to unseen reward functions (metrics)?*

As demonstrated in a recent work (Ma et al., 2025), reward hacking (Skalse et al., 2022) can significantly impair test-time scaling performance, where the model exploits flaws or ambiguities in the reward function to obtain high rewards. However, our method, EvoSearch, can mitigate

Table 1: Evaluation results across multiple metrics from both Vbench and VBench2.0.

| Methods | Dynamic | Semantic | Human Fidelity | Composition | Physics | Aesthetic | Average |
|---|---|---|---|---|---|---|---|
| Wan 1.3B | 13.18 | 16.83 | 82.98 | 38.08 | 64.44 | 64.01 | 46.59 |
| +Best of N | 15.38 ↑+2.2 | 13.67 ↓−3.16 | **87.58** ↑+4.6 | 44.71 ↑+6.63 | 56.10 ↓−8.34 | **64.84** ↑+0.83 | 47.04 ↑+0.45 |
| +Particle Sampling | 13.18 ↑+0.0 | 12.67 ↓−4.16 | 86.13 ↑+3.15 | 39.43 ↑+1.35 | 56.41 ↓−8.03 | 64.54 ↑+0.53 | 45.39 ↓−1.2 |
| +EvoSearch (Ours) | **16.48** ↑+3.3 | **15.51** ↓−1.32 | 86.84 ↑+3.86 | **51.57** ↑+13.49 | **57.5** ↓−6.9 | 64.35 ↑+0.34 | **48.71** ↑+2.12 |
| HunyuanVideo 13B | 8.79 | 16.11 | 90.28 | 47.89 | 56.10 | 66.31 | 47.58 |
| +Best of N | 6.59 ↓−2.2 | 12.84 ↓−3.27 | 91.31 ↑+1.03 | **50.53** ↑+2.64 | 47.62 ↓−8.48 | 66.28 ↓−0.03 | 45.86 ↓−1.72 |
| +Particle Sampling | 6.59 ↓−2.2 | 11.00 ↓−5.11 | 93.17 ↑+2.89 | 36.67 ↓−11.22 | 54.29 ↓−1.81 | 65.55 ↓−0.76 | 44.55 ↓−3.03 |
| +EvoSearch (Ours) | 7.69 ↓−1.1 | 14.92 ↓−1.19 | 94.63 ↑+4.35 | 51.37 ↑+3.48 | 61.54 ↑+5.44 | 66.75 ↑+0.44 | 49.48 ↑+1.90 |

Prompt: *A lion doing a handstand, balancing perfectly on its front paws while gazing confidently at the audience.*

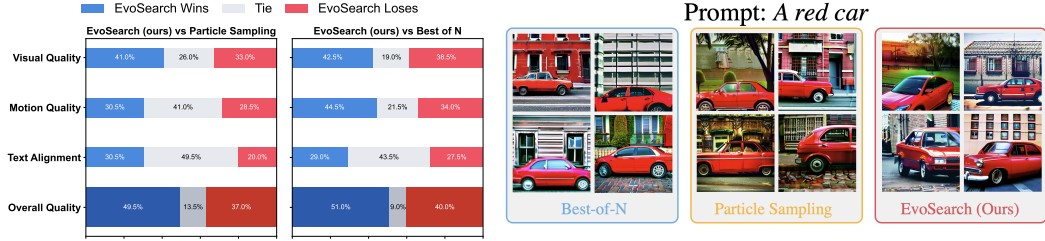

Best-of-N     Particle Sampling     EvoSearch (Ours)

Figure 8: A qualitative example showing that EvoSearch generates videos with superior visual quality, enhanced background consistency, and improved semantic alignment with the input text prompts.

the reward hacking problem to some extent since it maintains higher diversity through the search process, effectively capturing multimodal modes from target distributions. We evaluate the generation performance on unseen (out-of-distribution) metrics in Fig. 7, where ClipScore is used as the guidance reward. EvoSearch still showcases superior scalability and performance across different models and metrics. For o.o.d. metric Aesthetic, which is not aligned with ClipScore (as demonstrated in Fig. 8 of (Ma et al., 2025)), EvoSearch shows less performance degradation compared to particle sampling.

For video generation tasks, we include 9 different unseen metrics spanning 6 main categories to evaluate EvoSearch's generalizability to unseen rewards. From the results shown in Table 1, we observe that EvoSearch consistently gains more stable performance improvements compared with baselines. Notably, even for metrics that are not aligned with VideoReward (e.g., Semantic), EvoSearch maintains robust performance with minimal degradation. For the physics metric on HunyuanVideo, EvoSearch even achieves distinctive performance improvements while both best-of-N and particle sampling exhibit significant degradation.

Figure 9: Human evaluation results.

Prompt: *A red car*

Figure 10: For the same prompt, EvoSearch generates more visually diverse images.

**Question 4.** *How does EvoSearch perform under human evaluation?*

To validate EvoSearch's alignment with human preferences, we conduct a comprehensive human evaluation study employing professional annotators. The assessment focused on four key dimensions: Visual Quality, Motion Quality, Text Alignment, and Overall Quality. As illustrated in Fig. 9, EvoSearch achieves higher win rates compared to baseline methods across all evaluation dimensions.

**Question 5.** *Can EvoSearch remains high diversity when maximizing guidance rewards?*

EvoSearch demonstrates superior capability in sampling diverse solutions through its continuous exploration of novel states during the search process. We randomly select 10 prompts from DrawBench, and generate 10 images per prompt using EvoSearch and baselines under $100\times$ scaled inference-time compute. After generation, we evaluate the quality of the generated images by ImageReward, and evaluate the diversity of these images by the $L_2$ distance between their corresponding hidden features extracted from the CLIP encoder. We observe in Table 2 that EvoSearch obtains the highest reward while achieving the highest diversity. Qualitative results in Fig. 10 further support this finding, revealing that EvoSearch generates text-aligned images with notably greater diversity in backgrounds and poses compared to baseline methods.

Table 2: Results of reward and diversity.

| Method | Reward | Diversity |
|---|---|---|
| Best of N | 0.16 | 0.62 |
| Particle Sampling | 0.13 | 0.94 |
| EvoSearch (Ours) | **0.18** | **1.34** |

**Question 6.** *Can EvoSearch enable smaller-scale model outperform larger-scale model?*

In image generation tasks, as illustrated in Fig. 5, SD2.1 achieves competitive performance compared to GPT4o with fewer than $5e3$ NFEs ($\approx 30$ seconds inference time). Qualitative results presented in Fig. 1 further demonstrate how EvoSearch enables smaller models to reach GPT4o's level through strategic inference-time scaling. For video generation tasks, we allocate $5\times$ inference computation to Wan 1.3B, ensuring equivalent inference time with Wan

Table 3: EvoSearch scales Wan 1.3B to have the same inference time as Wan 14B. Results are evaluated on 625 prompts from VBench and VBench2.0.

| Methods | VideoReward |
|---|---|
| Wan 14B | -1.24 |
| Wan 1.3B + EvoSearch (ours) | **-0.15** |

14B on identical GPUs. Results documented in Table 3 show that the Wan 1.3B model with EvoSearch achieves competitive performance to its $10\times$ larger counterpart, the Wan 14B model. These findings highlight the significant potential of test-time scaling as a complement to traditional training-time scaling laws for visual generative models, opening new avenues for future research.

## 6 DISCUSSIONS

In this work, we propose Evolutionary Search (EvoSearch), a novel, generalist and efficient test-time scaling framework for diffusion and flow models across image and video generation tasks. Through our proposed specialized evolutionary mechanisms, EvoSearch enables the generation of higher-quality samples iteratively by actively exploring new states along the denoising trajectory. Limitations and future work are discussed in Appendix C.

## REPRODUCIBILITY STATEMENT

To ensure reproducibility, we provide a detailed experimental setup and hyperparameters used during training and evaluation in Appendix A.1. Moreover, we provide our codebase at `https://anonymous.4open.science/r/EvoSearch`.

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

# A EXPERIMENTAL DETAILS

## A.1 IMPLEMENTATION DETAILS

### A.1.1 IMPLEMENTATION DETAILS OF EVOSEARCH

**Evolution schedule $\mathcal{T}$.** Evolution schedule $\mathcal{T}$ can be flexibly defined based on the available amount of inference-time compute. If the inference-time computation budget is sufficient, we can perform EvoSearch at more timesteps; otherwise, we can deploy EvoSearch at several timesteps. In our implementation, we set $\mathcal{T}$ to have uniform intervals.

**Population size schedule $\mathcal{K}$.** Population size schedule is defined as $\mathcal{K} = \{k_{\text{start}}, k_T, \cdots, k_j, \cdots, k_n\}$. $\mathcal{K}$ can be flexibly defined based on the available amount of inference-time compute. We can increase population size as inference-time computation increases. In our implementation, we assign $2\times$ larger population size at the first generation of EvoSearch, while keeping the population size at the remaining generations the same. This means that $k_{\text{start}}$ is twice as large as the other population sizes.

**Stable Diffusion 2.1.** We set the guidance scale as 5.5, and set the resolution size as $512 \times 512$. We employ the DDIM scheduler from the diffusers library (von Platen et al., 2022) for inference. We set the mutation rate $\beta = 0.3$, with $\sigma_t$ following the default DDIM configurations.

**Flux.1-dev.** We set the guidance scale as 5.5, and set the resolution size as $512 \times 512$. We employ the *sde-dpmsolver++* sampler in *FlowDPMSolverMultistepScheduler* (von Platen et al., 2022) for inference in SDE process. We set the mutation rate $\beta = 0.3$, with $\sigma_t$ following the default sde-dpmsolver configurations.

**Wan.** Following the official codes (Wang et al., 2025), we set the resolution size as $832 \times 480$, with a video consists of 33 frames. We set the guidance scale as 5.0. For transforming the ODE denoising process in Wan to SDE process, we leverage the *sde-dpmsolver++* sampler in *FlowDPMSolverMultistepScheduler* (von Platen et al., 2022) for inference.

**Hunyuan.** Following the official implementation (Kong et al., 2024), we set the resolution size as $544 \times 960$ to ensure the generation quality, with a video consisting of 33 frames. The guidance scale is set at 1.0 as suggested, and the embedded guidance scale is 6.0. For transforming the ODE denoising process in Wan to SDE process, we leverage the *sde-dpmsolver++* sampler in *FlowDPMSolverMultistepScheduler* (von Platen et al., 2022) for inference. To save computation for a large number of experiments conducted in this paper, we set the inference steps to 30.

We refer to the pseudocodes of EvoSearch in Alg. 1 and Alg. 2. At the beginning of EvoSearch, we denote the size of randomly sampled Gaussian noises as $k_{\text{start}}$. The implementation of EvoSearch is provided in the supplementary material, ensuring reproducibility.

---

**Algorithm 1** Overview of EvoSearch

1: **Input:** Pre-trained model $p_\theta$, population size schedule $\mathcal{K} = \{k_{\text{start}}, k_T, \cdots, k_j, \cdots, k_n\}$, evolution schedule $\mathcal{T} = \{T, \cdots, t_j, \cdots, t_n\}$
2: Initialize population list $\mathcal{P} = [\phi \text{ for } \_ \text{ in } \mathcal{T}]$.
3: Initialize reward list $\mathcal{R} = [\phi \text{ for } \_ \text{ in } \mathcal{T}]$
4: Sample initial Gaussian noises $\boldsymbol{x}_T$ with population size $k_{\text{start}}$
5: Initialize generation $g = 0$
6: **for** $t = T, T-1, \cdots, 1$ **do**
7:    **if** $t$ in $\mathcal{T}$ **then**
8:      $\boldsymbol{x}_t, \mathcal{P}, \mathcal{R} = evosearch\_at\_denoising\_states(p_\theta, \boldsymbol{x}_t, \mathcal{P}, \mathcal{R}, \mathcal{T}, \mathcal{K}, g)$    // Alg 2
9:      $g \leftarrow g + 1$
     Evolutionary generation process
10:    **end if**
11:    $\boldsymbol{x}_{t-1} = \text{denoise}(p_\theta, \boldsymbol{x}_t, t)$
   Standard denoising process
12: **end for**

---

---

**Algorithm 2** EvoSearch at Denoising States

---

1: **Input:** Pre-trained model $p_\theta$, starting states $\boldsymbol{x}_{t'}$, population list $\mathcal{P}$, reward list $\mathcal{R}$, evolution schedule $\mathcal{T} = \{T, \cdots, t_j, \cdots, t_n\}$, population size schedule $\mathcal{K} = \{k_T, \cdots, k_j, \cdots, k_n\}$, generation $g$, elites size $m$.
2: Set idx = g
3: Set population size $k = \mathcal{K}[g+1]$
4: **for** $t = t', t'-1, \cdots, 1$ **do**
5:     **if** $t$ in $\mathcal{T}$ **then**
6:         $\mathcal{P}[\text{idx}] = \text{cat}(\mathcal{P}[\text{idx}], \boldsymbol{x}_t)$
7:         idx $\leftarrow$ idx + 1
8:     **end if**
9:     $\boldsymbol{x}_{t-1} = \text{denoise}(\boldsymbol{x}_t, t)$
10: **end for**
11: Calculate rewards $r$ via fully denoised $\boldsymbol{x}_0$ in Eq. equation 2
12: **for** $i = g, \cdots, \text{len}(\mathcal{R}) - 1$ **do**
13:     $\mathcal{R}[i] = \text{cat}(\mathcal{R}[i], r)$   // Compute a single reward per $\boldsymbol{x}_0$
14: **end for**
15: Select elites $e = \mathcal{P}[g]\,[\text{topk}(\mathcal{R}[g], m)]$
16: Select $k - m$ parents $p$ from $\mathcal{P}[g]$ via tournament selection (Goldberg & Deb, 1991)
17: **if** g=0 **then**
18:     Mutate parents $p = \sqrt{1 - \beta^2} \times p + \varepsilon \times \beta, \;\; \varepsilon \sim \mathcal{N}(0, I)$
19: **else**
20:     Mutate parents $p = p + \sigma_t \times \varepsilon, \;\; \varepsilon \sim \mathcal{N}(0, I)$
    // $\sigma_t$ is the diffusion coefficient in the SDE denoising process
21: **end if**
22: Get children $c \leftarrow \text{cat}(e, p)$
23: **Output:** Children $c, \mathcal{P}, \mathcal{R}$

---

### A.1.2 IMPLEMENTATION DETAILS OF BASELINES

**Best of N.** Best of N generates a batch of N candidate samples (images or videos), from which the highest-quality sample is selected according to a predefined guidance reward function. In practice, we use the same guidance reward for EvoSearch and all baselines to ensure fair comparison.

**Particle Sampling.** Particle-based sampling methods have demonstrated significant effectiveness in enhancing the generative performance of diffusion models during inference. For our implementation, we leverage the generalist particle-based sampling framework proposed by (Singhal et al., 2025a), utilizing their publicly available codebase. Their approach introduces a flexible methodology that accommodates diverse potential functions, sampling algorithms, and reward models, leading to improved performance across a broad spectrum of text-to-image generation tasks. We adopt the *Max* potential schedule for resampling at intermediate states, which empirically demonstrated superior performance in the original study. Other hyperparameters, such as the *resampling interval*, are carefully tuned to establish a robust baseline performance.

### A.2 THEORETICAL ANALYSIS OF INTERMEDIATE STATE MUTATION

**Definition 1** (SDE Denoising Process). *Let $\{x_t\}_{t=0}^T$ denote the state sequence in a stochastic differential equation (SDE) denoising process. The reverse-time transition from $x_t$ to $x_{t-1}$ follows:*

$$x_{t-1}^{parent} = x_t - f_t(x_t) + \sigma_t \varepsilon_1, \quad \varepsilon_1 \sim \mathcal{N}(0, \mathbf{I}) \tag{5}$$

*where $f_t : \mathbb{R}^d \to \mathbb{R}^d$ is a drift function, $\sigma_t > 0$ is the diffusion coefficient at timestep $t$, and $\varepsilon_1$ is standard Gaussian noise.*

**Theorem 1** (Validity of Mutation Scheme). *The proposed mutation operator $\mathcal{M} : \mathbb{R}^d \to \mathbb{R}^d$ defined as*

$$\mathcal{M}(x_{t-1}^{parent}) = x_{t-1}^{parent} + \sigma_t \varepsilon_2, \quad \varepsilon_2 \sim \mathcal{N}(0, \mathbf{I}) \tag{6}$$

*satisfies the following properties:*

    *1. Well-definedness: $\mathcal{M}$ generates valid state transitions.*

*2. SDE consistency: Mutated states adhere to the reverse-time SDE dynamics.*

*Proof.* Let $x_{t-1}^{\text{child}} = \mathcal{M}(x_{t-1}^{\text{parent}})$. Substituting Definition 1 into the mutation operator:

$$x_{t-1}^{\text{child}} = [x_t - f_t(x_t) + \sigma_t \varepsilon_1] + \sigma_t \varepsilon_2$$
$$= x_t - f_t(x_t) + \sigma_t(\varepsilon_1 + \varepsilon_2). \tag{7}$$

Since $\varepsilon_1 \sim \mathcal{N}(0, \mathbf{I})$ and $\varepsilon_2 \sim \mathcal{N}(0, \mathbf{I})$ are independent, their sum follows:

$$\varepsilon \triangleq \varepsilon_1 + \varepsilon_2 \sim \mathcal{N}(0, 2\mathbf{I}). \tag{8}$$

By substitution, we have:

$$x_{t-1}^{\text{child}} = x_t - f_t(x_t) + \sigma_t \varepsilon. \tag{9}$$

Therefore, the marginal distribution $p_t(x_{t-1}^{\text{child}})$ after mutation remains Gaussian:

$$p_t(x_{t-1}^{\text{child}}) = \mathbb{E}_{x_t}\left[\mathcal{N}\left(x_{t-1}\,;\, x_t - f_t(x_t),\, 2\sigma_t^2\mathbf{I}\right)\right]. \tag{10}$$

This matches the SDE transition form with a modified diffusion coefficient $\sqrt{2}\sigma_t$, which expands the exploration space without hindering the denoising process, as the diffusion coefficient can be chosen freely within the stochastic interpolant framework (Ma et al., 2024; Albergo et al., 2023). $\square$

### A.3 EVALUATION METRICS

**Image Evaluation Metrics.** (i) **ImageReward** is a text-to-image human preference reward model (Xu et al., 2023), which takes an image and its corresponding prompt as inputs and outputs a preference score. (ii) **CLIPScore** is a reference-free evaluation metric derived from the CLIP model (Hessel et al., 2021), which aligns visual and textual embeddings in a shared latent space. By computing the cosine similarity between an image embedding and its associated text prompt embedding, CLIPScore quantifies semantic coherence without requiring ground-truth images. (iii) **HPSv2** is a preference prediction model that reflects human perceptual preferences for text-to-image generation (Wu et al., 2023b). (iv) **Aesthetic** quantifies the visual appeal of images, often independent of text prompts (Schuhmann et al., 2022).

**Video Evaluation Metrics.** (i) **Dynamic** evaluates a model's ability to follow complex prompts and simulate dynamic changes (i.e., color, size, lightness, and material). This evaluation metric includes prompts of *Dynamic Attribute* form VBench2.0. Scores are calculated following the original codes (Zheng et al., 2025). (ii) **Semantic** evaluates the model's ability to follow long prompts, which involve at least 150 words. This evaluation metric includes the prompts of *Complex Plot* and *Complex Landscape* from VBench2.0. (iii) **Human Fidelity** evaluates both the structural correctness and temporal consistency of human figures in generated videos. This evaluation metric includes the prompts of *Human Anatomy*, *Human Clothes*, and *Human Identities* from VBench2.0. (iv) **Composition** evaluates the model's ability to generate complex, impossible compositions beyond real-world constraints. This evaluation metric includes the prompts of *Composition* from VBench 2.0. (v) **Physics** evaluates whether models follow basic real-world physical principles (e.g., gravity). This evaluation metric includes the prompts of *Mechanics* from VBench2.0. (vi) **Aesthetic** evaluates the aesthetic values perceived by humans towards each video frame using the LAION aesthetic predictor (Schuhmann et al., 2022). This evaluation metric includes the prompts of *Aesthetic Quality* from VBench.

## B ADDITIONAL EXPERIMENTAL RESULTS

### B.1 ABLATION ON POPULATION SIZE SCHEDULE

To ablate the effect of population size schedules under the same inference-time computation budget, we set different population size schedules for the Stable Diffusion 2.1 model with approximately $140 \times 50$ inference-time NFEs. Here, 50 is the length of the denoising steps for each generation. We report the DrawBench results in Fig. 11. We observe that different population size schedules perform similarly with little reward difference. The most significant factor is the value of $k_{\text{start}}$, which represents the population size of the initial Gaussian noises. A larger value of $k_{\text{start}}$ benefits a strong initialization for the subsequent search process, while a small value of $k_{\text{start}}$ would affect the performance a lot.

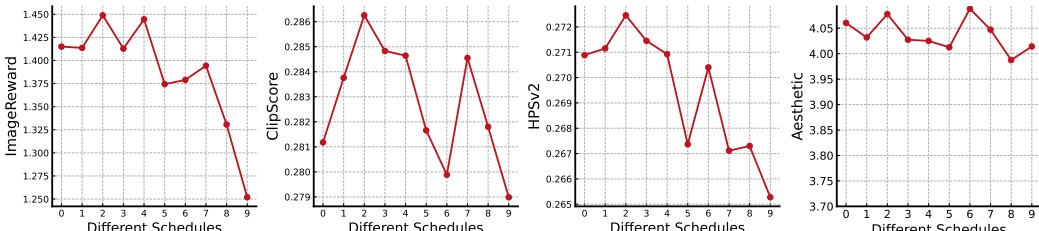

Figure 11: Ablation study on the population size schedule $\mathcal{K}$. We denote the population size schedule $\mathcal{K} = \{k_{\text{start}}, k_T, \cdots, k_j, \cdots, k_n\}$, where $k_{\text{start}}$ is the size of the initial sampled Gaussian noises. We use Stable Diffusion 2.1 to conduct EvoSearch on DrawBench, employing ImageReward as the guidance reward function during search, and the denoising step is 50. From left to right of the x-axis, the population size schedule $\mathcal{K}$ is configured as: 0) $\{60, 40, 50\}$; 1) $\{70, 30, 50\}$; 2) $\{80, 20, 50\}$; 3) $\{62, 62, 20\}$; 4)$\{58, 58, 30\}$; 5) $\{54, 54, 40\}$; 6) $\{46, 46, 60\}$;7) $\{40, 60, 50\}$; 8) $\{30, 70, 50\}$; 9) $\{20, 80, 50\}$, where we maintain the evolution schedule as $\{50, 40\}$.

### B.1.1 Ablation on Evolution Schedule

We further ablate the effect of the evolution schedule. From the results shown in Fig. 12, we find that the evolution schedule $\mathcal{T}$ exhibits less significant influence compared to the population size schedule $\mathcal{K}$. Our analysis demonstrates that an evolution schedule with uniform intervals yields superior performance. Additionally, larger initial population sizes $k_{\text{start}}$ help increase the performance.

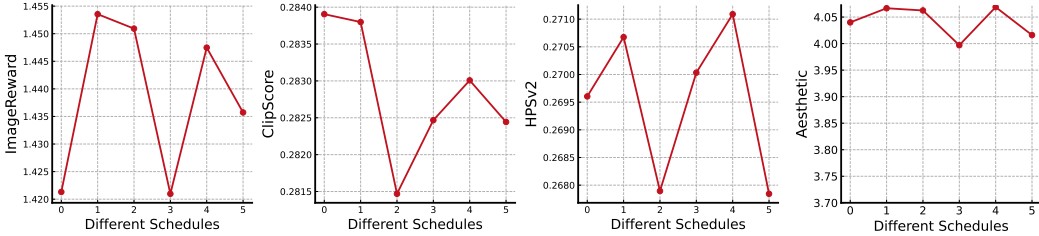

Figure 12: Ablation study on the evolution schedule $\mathcal{T}$. We use Stable Diffusion 2.1 to conduct EvoSearch on the DrawBench, employing ImageReward as the guidance reward function during search. We denote the evolution schedule $\mathcal{T} = \{T, \cdots, t_m, \cdots, t_n\}$. From left to right of the x-axis, the evolution schedule is 0) $\{50, 30\}$; 1) $\{50, 20\}$; 2) $\{50, 10\}$; 3) $\{50, 30\}$; 4) $\{50, 20\}$; 5) $\{50, 10\}$. To keep the same test-time scaling computation budget across different evolution schedules, each population size schedule is adjusted as 0) $\{60, 50, 50\}$; 1) $\{70, 50, 50\}$; 2) $\{80, 50, 50\}$; 3) $\{55, 55, 50\}$; 4) $\{60, 60, 50\}$; 5) $\{75, 75, 50\}$.

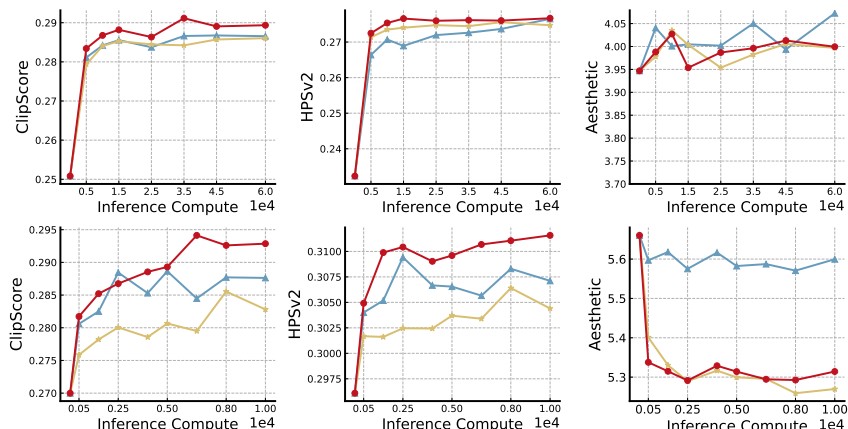

Figure 13: EvoSearch can generalize to unseen metrics, where ImageReward is set as the guidance reward function during search. *Top row*: DrawBench results on SD2.1. *Bottom row*: DrawBench results on Flux.1-dev.

## B.2 WALL-CLOCK TIME ANALYSIS

We show the wall-clock time required for different methods in Fig. 14. We observe that EvoSearch achieves superior performance compared with baselines given the same wall-clock time, demonstrating the efficiency and effectiveness of our proposed method. Both particle sampling and Best-of-N can rapidly fall into a plateau, while EvoSearch continues to improve the base models' performance with increased computation. The efficiency of EvoSearch lies in its progressive evolution framework: (1) EvoSearch only needs a single reward evaluation at the end of each evolution generation, while particle sampling requires multiple reward computations at intermediate steps per particle. (2) EvoSearch uniquely caches all intermediate samples at evolution timesteps $t \in \mathcal{T}$, creating a rich pool of parent candidates for subsequent evolution generations. This mechanism avoids repeatedly denoising from $x_T$ across each evolution branch and eliminates redundant denoising computations.

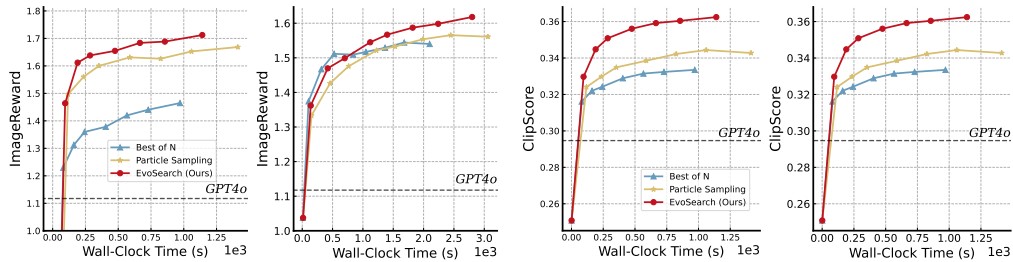

(a) ImageReward as target guidance. *Left:* SD2.1; *Right:* Flux.1-dev

(b) ClipScore as target guidance. *Left:* SD2.1; *Right:* Flux.1-dev

Figure 14: We compare EvoSearch with baselines based on wall-clock time per prompt. We record the time for different methods on the same hardware and GPU card to ensure fairness.

## B.3 RESULTS ON GENEVAL AND DPGBENCH

To further showcase the effects of EvoSearch generalizing to different evaluation metrics and benchmarks, we compare EvoSearch with baselines on GenEval (Ghosh et al., 2023) and DPGBench (Hu et al., 2024), which include fine-grained assessment across multiple dimensions (e.g., color, count) and carefully designed prompts. As shown in Table 4, EvoSearch improves SD2.1's score on GenEval by 83.6%, finally surpassing GPT4o. The results provided in Table 5 demonstrate that EvoSearch continues to outperform all the baselines on DPGBench, and we find that Flux.1-dev with EvoSearch can also surpass GPT4o.

Table 4: Following the official evaluation pipeline of GenEval (Ghosh et al., 2023), we compare EvoSearch to baselines with $200\times$ NFEs available at test-time. SD2.1 is used as the base model. We employ the scores defined in GenEval as the guidance rewards during search.

| Methods | Geneval Score |
|---|---|
| SD2.1 | 0.50 |
| GPT4o | 0.84 |
| **EvoSearch w/ SD2.1** | **0.92** |
| Particle Sampling w/ SD2.1 | 0.86 |
| Best of N w/ SD2.1 | 0.83 |

Table 5: We evaluate EvoSearch and the baselines with $10\times$ NFEs allocated at test time on the 1065 prompts provided by DPGBench (Hu et al., 2024), leveraging the pre-defined DPG score as the guidance reward during search. Flux.1-dev is used as the base model, with 50 denoising steps per generation.

| Methods | DPG Score |
|---|---|
| Flux.1-dev | 83.84 |
| GPT4o | 85.15 |
| **EvoSearch w/ Flux.1-dev** | **93.51** |
| Particle Sampling w/ Flux.1-dev | 89.32 |
| Best of N w/ Flux.1-dev | 90.06 |

### B.4 QUALITATIVE RESULTS

We present extensive qualitative results for both image and video generation as follows. The images of GPT4o are generated by the OpenAI API following the default configuration. To ensure fair comparison, the prompts given to GPT4o remain the same as those of other models.

#### B.4.1 RESULTS FOR IMAGE GENERATION

Please refer to Fig. 15, Fig. 16, and Fig. 17 for comparison between EvoSearch and baselines. These examples clearly demonstrate that EvoSearch significantly enhances image generation performance while requiring lower computational resources.

#### B.4.2 RESULTS FOR VIDEO GENERATION

Please refer to Fig. 18, Fig. 19, Fig. 20, and Fig. 21 for comparison between EvoSearch and baselines in the context of video generation. We find that EvoSearch outperforms all the baselines with higher efficacy and efficiency. Please refer to Fig. 22, Fig. 23, Fig. 24, Fig. 25, Fig. 26, Fig. 27, and Fig. 28 for comparison between Wan14B and Wan1.3B enhanced with EvoSearch. For more details, please visit the anonymous website `evosearch.github.io`. The results demonstrate that by increasing the test-time computation budget of Wan1.3B to match the inference latency of Wan14B, the smaller model outperforms its $10\times$ larger counterpart across a diverse range of input prompts.

## C DISCUSSIONS

**Limitations and Future Work.** EvoSearch has demonstrated significant effectiveness in exploring high-reward regions of novel states, which opens promising directions for future research. The exploration ability of EvoSearch relies on the strength of the mutation rate $\beta$ and $\sigma_t$. A higher mutation rate will effectively expand the search space to find high-quality candidates, while a low mutation rate can restrict the exploration space, which represents a trade-off. In addition, we rely on Gaussian noise to mutate the selected parents. While this approach provides robust exploration across diverse image and video generation tasks, developing more informative mutation strategies with prior knowledge can further improve the search efficiency. The inherent complexity of interpreting denoising states makes it an interesting open research question. Our findings also suggest promising future directions in understanding the shared structure between "golden" noise

and "golden" intermediate denoising states, which may provide valuable insights for future test-time scaling research.

**Broader Impacts.** This work proposes a novel test-time scaling method, called EvoSearch, for image and video generation tasks across both diffusion-based and flow-based models. EvoSearch draws inspiration from biological evolution (Ao, 2005), which significantly improves both the quality and diversity of generated samples through enhanced exploration during the search process. Our proposed method is promising to provide insights for test-time scaling in other areas, like large language models (LLMs). Therefore, our proposed method can further enhance the research of test-time scaling and inference-time alignment in the general area of machine learning. No significant negative broader impacts were identified that warrant specific emphasis in this paper.

## D    COMPARISON AND DISCUSSION WITH GRADIENT-BASED METHODS

Compared with gradient-based methods like ReNO (Eyring et al., 2024) and D-Flow (Ben-Hamu et al., 2024), our proposed method, EvoSearch, has the following advantages: (1) **Universality**: Not all reward functions are differentiable (e.g., rewards from proprietary APIs, discrete metrics, or human feedback). A gradient-free approach makes EvoSearch universally applicable. (2) **Memory & Efficiency**: Backpropagating gradients through the diffusion process is memory-intensive and computationally expensive. Training-free methods like test-time scaling are significantly cheaper, allowing us to explore a wider search space within the same wall-clock time. (3) **Avoiding Local Optima**: Gradient ascent on noise latent space is prone to getting stuck in local optima or generating "adversarial" examples (e.g., high reward score but poor visual quality), which is a well-known problem in the literature. Our proposed EvoSearch, which is based on a gradient-free evolutionary algorithm, is better suited for non-convex landscapes as it maintains population diversity explicitly.

We compare EvoSearch against ReNO (Eyring et al., 2024), a representative gradient-based method. From the results shown in Table 6, we observe that EvoSearch significantly outperforms ReNO given the same wall-clock time. ReNO suffers from diminishing returns, improving only slightly (0.65 → 0.68) even when computation is increased by 10×. In contrast, EvoSearch effectively converts increased compute into quality, reaching a 0.77 score.

| Methods | Computation / Time | GenEval Score |
|---|---|---|
| SDXL-Turbo (Base) | - | 0.54 |
| +ReNO | 50 iters / 33s | 0.65 |
| +EVOSearch | 190 NFEs / 33s | **0.71** |
| +ReNO | 500 iters / 330s | 0.68 |
| +EVOSearch | 1000 NFEs / 172s | **0.77** |

Table 6: Both ReNO and EvoSearch employ the combination of the reward models (HPS, ImageReward, CLIP, PickScore) as the guidance, with SDXL-Turbo as the base model. We compare based on Wall-Clock Time to ensure a fair assessment of practical utility.

## E    COMPARISON WITH ROLLOVER BUDGET FORCING (RBF)

To further demonstrate the advantages of EvoSearch, we have added Rollover Budget Forcing (RBF) (Kim et al., 2025a) for comparison. Following RBF's official implementation, we employ Flux as the base model with ImageReward as the guidance. We fix the total number of function evaluations (NFEs) to 500 and set the number of denoising steps to 10. We evaluate both EvoSearch and RBF on DrawBench. We provide the results in Table 7. While the numerical margins may appear modest, we emphasize that on the DrawBench scale, a consistent improvement across 200 prompts on three different metrics (including both target and unseen metrics) confirms that EvoSearch generates strictly superior samples.

| Methods | ImageReward (target) | ClipScore (o.o.d.) | HPSv2(o.o.d.) |
|---------|---------------------|--------------------|----------------|
| RBF | 1.38 | 0.281 | 0.305 |
| EvoSearch | **1.41** | **0.284** | **0.310** |

Table 7: EvoSearch outperforms RBF given the same test-time NFEs, including both target reward and unseen reward.

## F  THE USE OF LARGE LANGUAGE MODELS (LLMS)

In compliance with ICLR 2026 policies on large language model usage, we disclose that LLMs are mainly used for writing polish in this work. We utilized LLMs to polish the paper's writing at the syntactic and grammatical levels. All LLM-generated content has undergone thorough human review and verification to ensure accuracy, appropriateness, and compliance with academic standards.

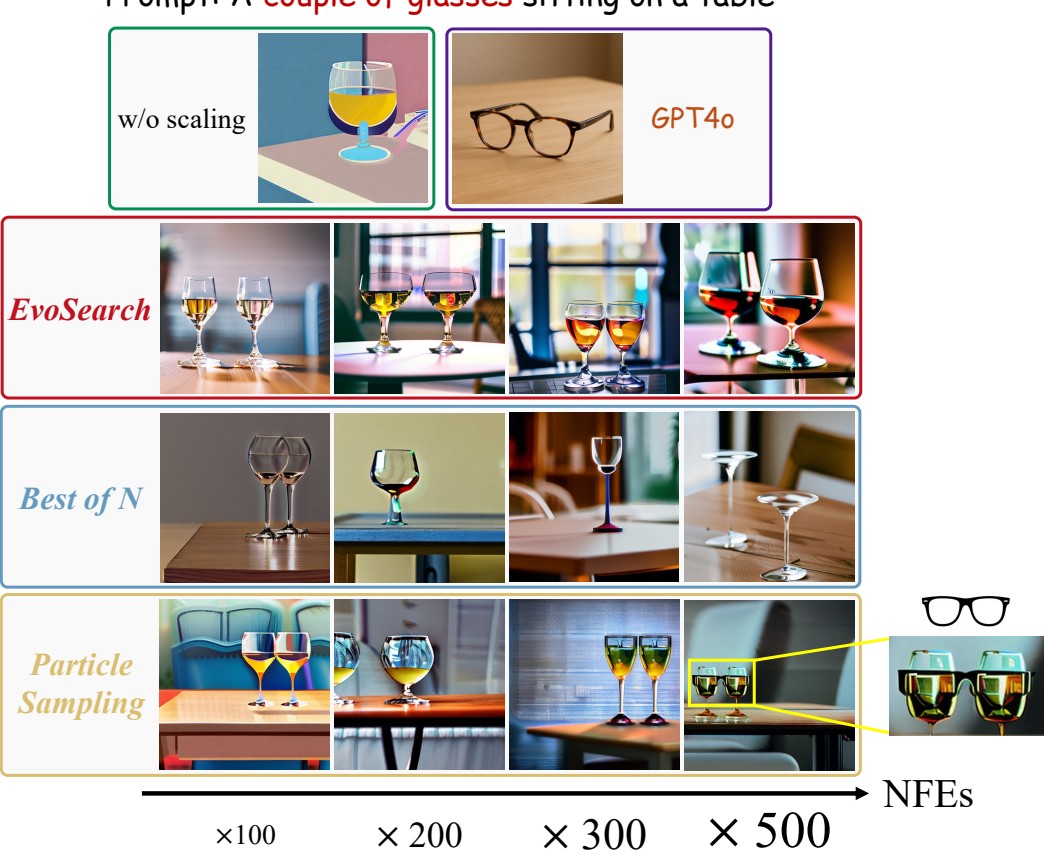

Figure 15: Comparative analysis of test-time scaling methods for Stable Diffusion 2.1. **EvoSearch** demonstrates consistent improvements in image quality and text-prompt alignment as NFEs increase, achieving accurate interpretations of the challenging prompt with high computational efficiency. In contrast, **Best-of-N** fails to produce semantically correct results even with increased NFEs, while **Particle Sampling** introduces semantic ambiguity at higher NFEs (e.g., confusing wine glasses and eyeglasses). Notably, **EvoSearch** further enables SD2.1 to outperform **GPT4o**.

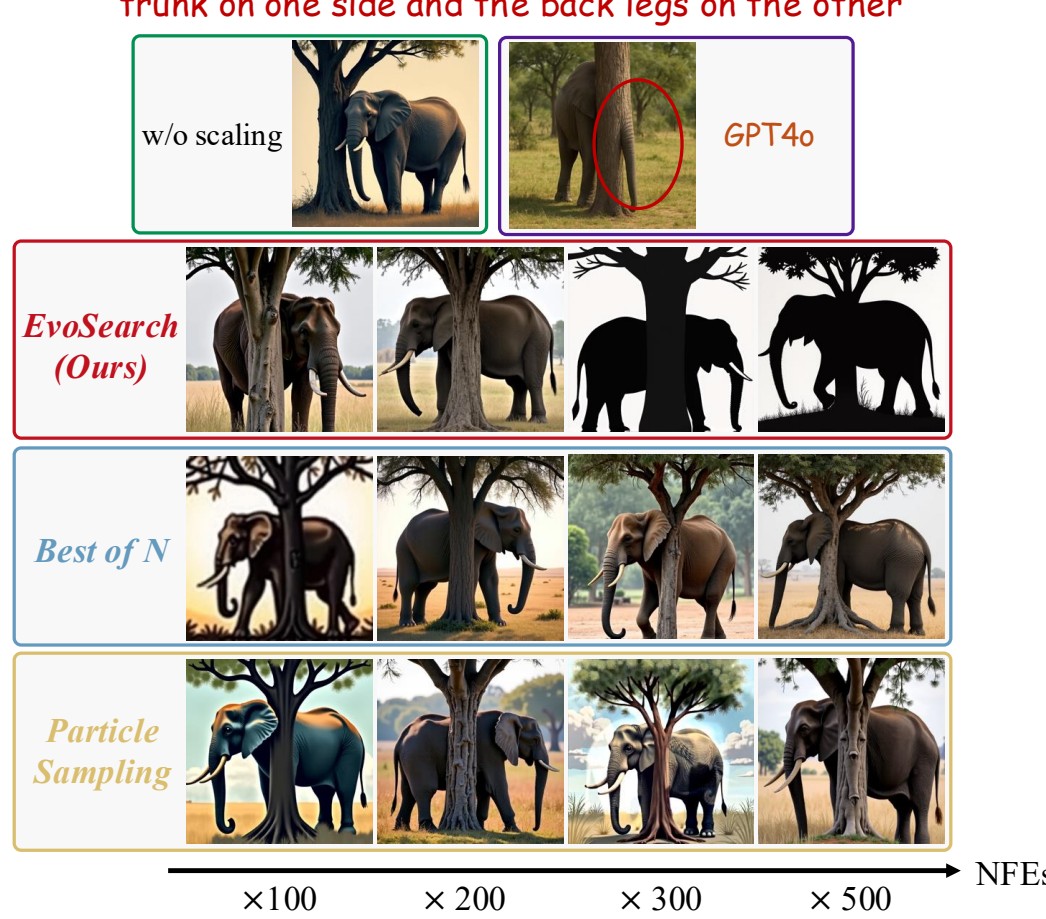

Figure 16: Results of test-time scaling for Flux.1-dev. **EvoSearch** demonstrates significant exploration ability, enabling the generation of images with diverse styles, while both **Best-of-N** and **Particle Sampling** generate images with reduced diversity.

Prompt: A laptop on top of a teddy bear

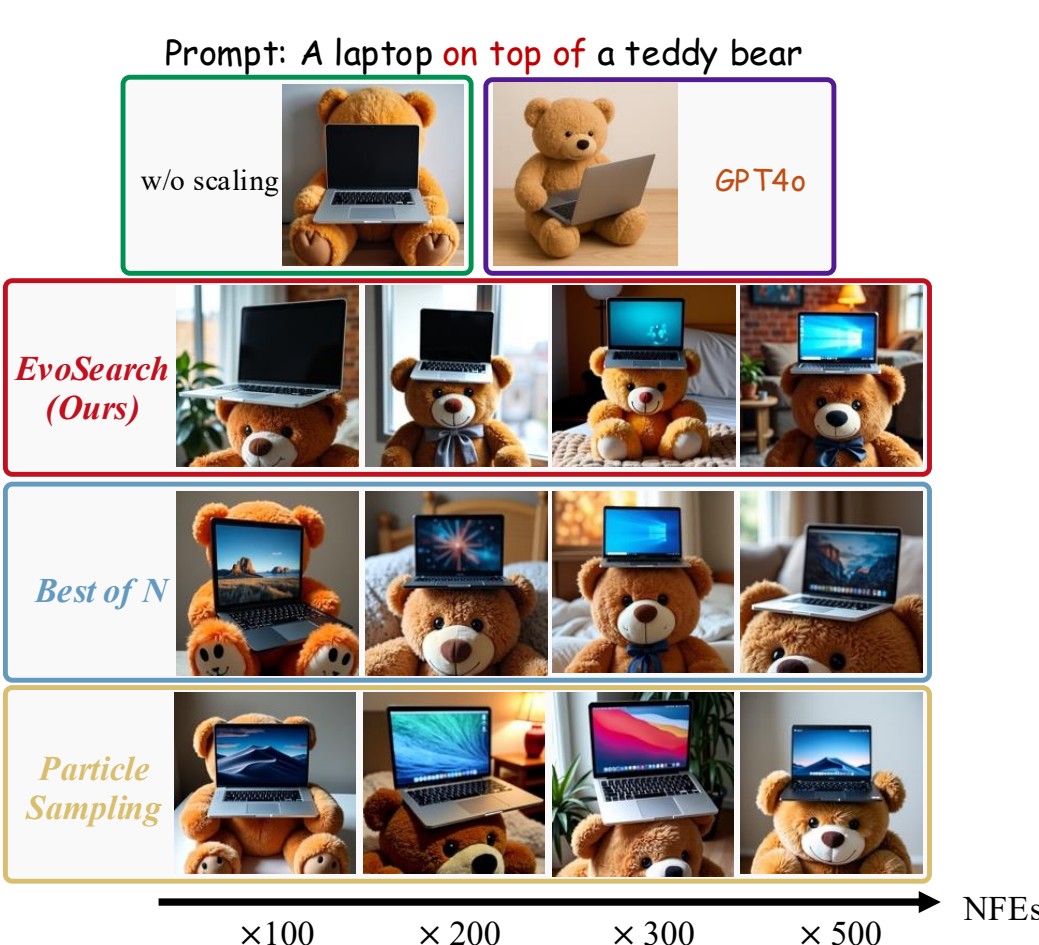

Figure 17: Results of test-time scaling for Flux.1-dev. **EvoSearch** can even achieve accurate spatial relationship interpretation with only $10\times$ scaled computation budget, while consistently improving image quality through higher NFEs.

Prompt: A spider with the body of a rabbit, scurrying across the ground with immense speed

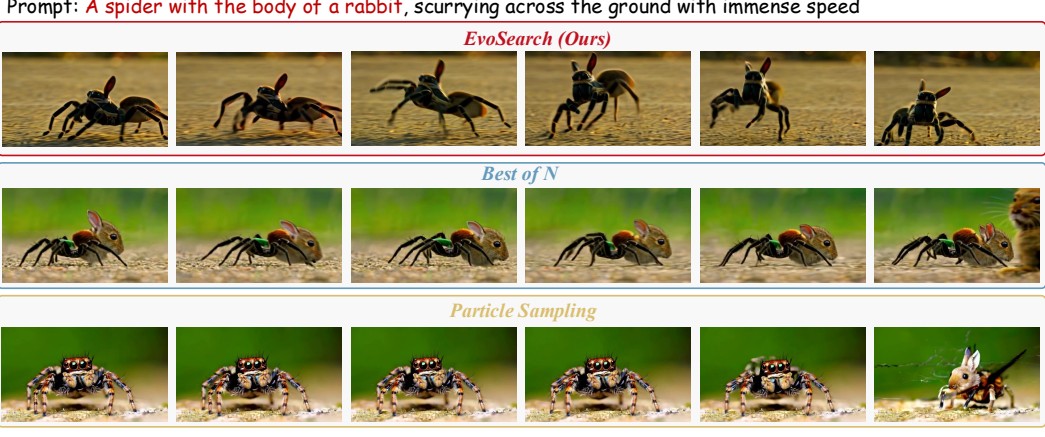

Figure 18: Results of test-time scaling for Hunyuan 13B. The denoising step is 30, and we scale up the test-time computation by $20\times$. Only **EvoSearch** generates high-quality video aligned closely with the text prompt.

Prompt: A cat is on the right of a rock, then the cat runs to the left of the rock

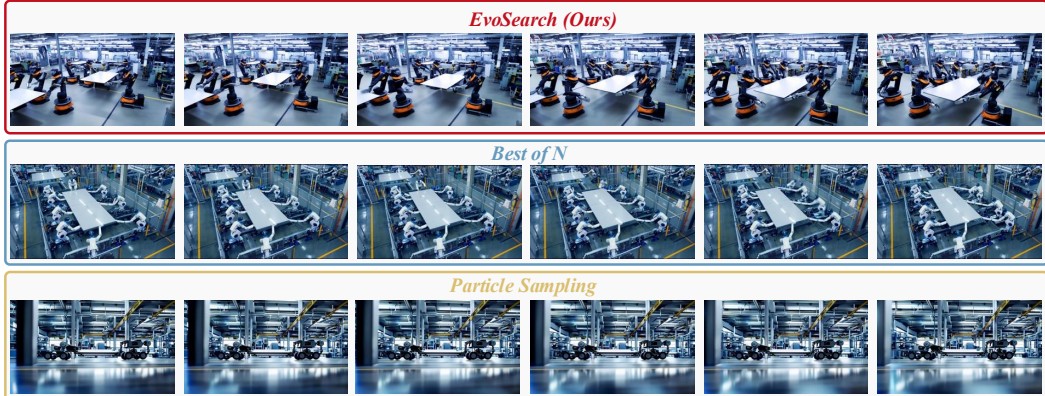

Figure 19: Results of test-time scaling for Hunyuan 13B. The denoising step is 30, and we scale up the test-time computation by $20\times$. **EvoSearch** successfully follows the text prompt while both **Best-of-N** and **Particle Sampling** fail.

Prompt: Several robots coordinate to move a large object across a factory floor. The camera captures the synchronized movements of the robots from a bird's-eye view, showing their precise coordination. The shot then shifts to ground level, focusing on the smooth, synchronized actions of the robots as they work together

Figure 20: Results of test-time scaling for Hunyuan 13B. The denoising step is 30, and we scale up the test-time computation by $20\times$. **EvoSearch** demonstrates superior text alignment and higher-quality generation compared to baselines.

Prompt: Two cars collide at an intersection.

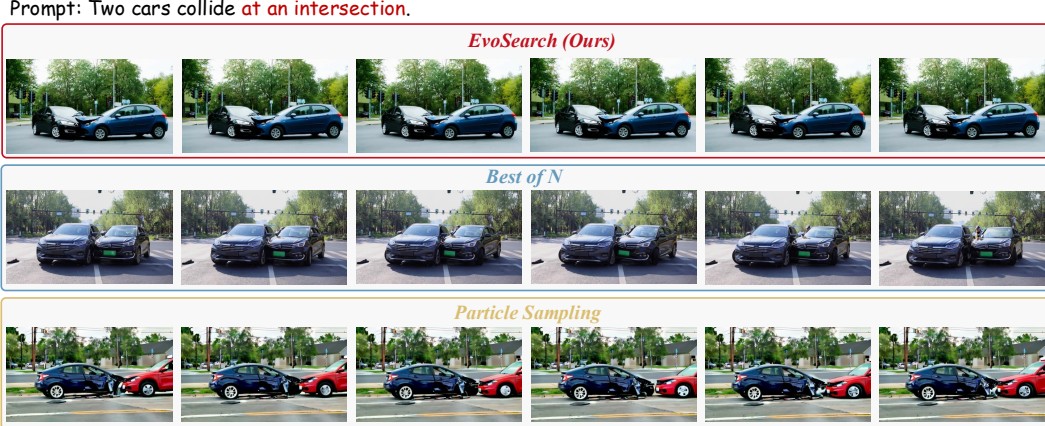

Figure 21: Results of test-time scaling for Hunyuan 13B. The denoising step is 30, and we scale up the test-time computation by $20\times$. The video generated by **EvoSearch** demonstrates better image quality and text alignment.

Prompt: An owl with the body of a tiger, prowling the night skies with sharp talons.

**Wan14B**

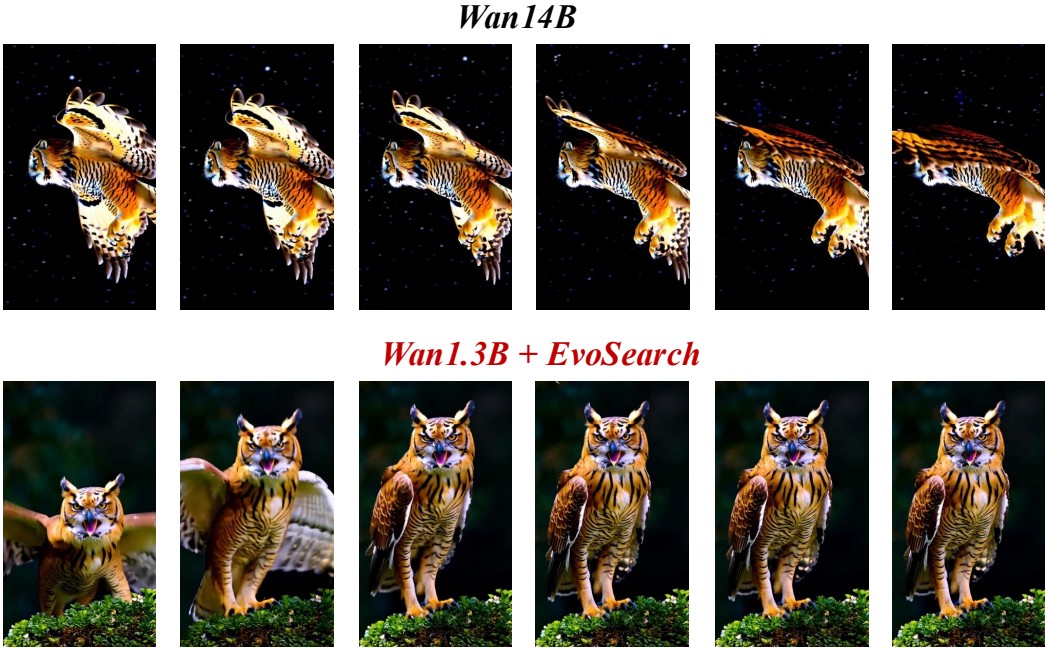

**Wan1.3B + EvoSearch**

Figure 22: We scale up the test-time computation of Wan1.3B by $5\times$, ensuring equivalent inference times between Wan14B and **Wan1.3B+EvoSearch**. Qualitative results demonstrate that **EvoSearch** enables Wan1.3B to outperform Wan14B, its $10\times$ larger counterpart.

Prompt: A cheetah doing yoga poses, stretching out its limbs with remarkable flexibility and focus

### *Wan14B*

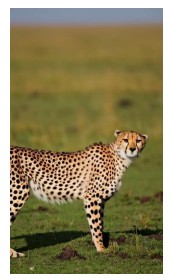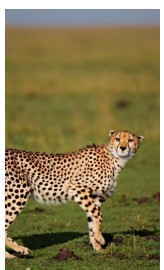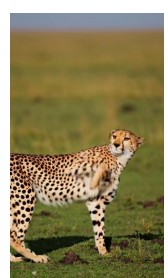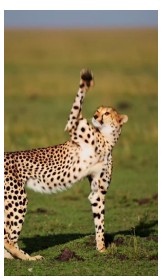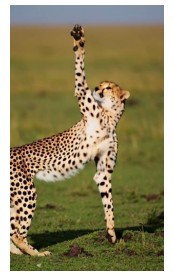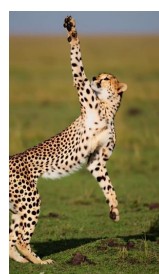

### *Wan1.3B + EvoSearch*

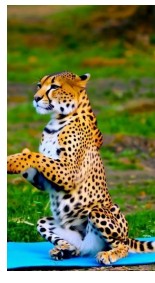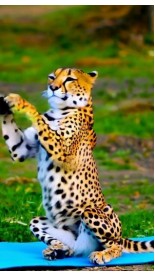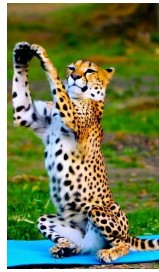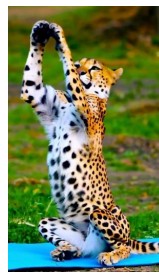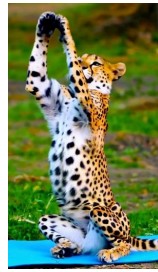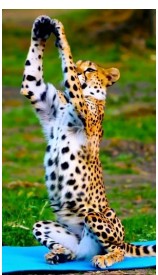

Figure 23: We scale up the test-time computation of Wan1.3B by $5\times$, ensuring equivalent inference times between Wan14B and **Wan1.3B+EvoSearch**. **EvoSearch** enables smaller models to achieve not only competitive but superior performance compared to their larger counterparts.

Prompt: A kite and a balloon flying side by side, each drifting gracefully in the wind.

### *Wan14B*

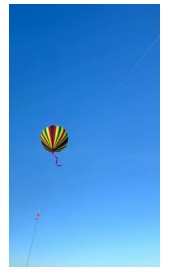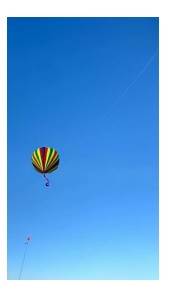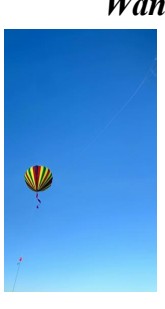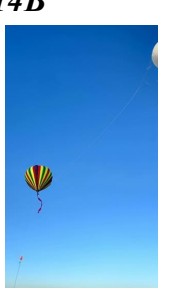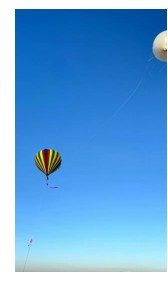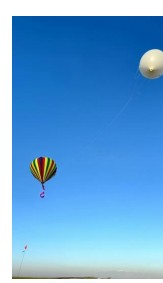

### *Wan1.3B + EvoSearch*

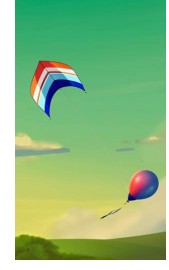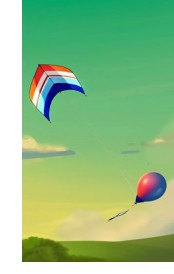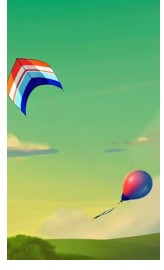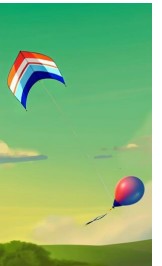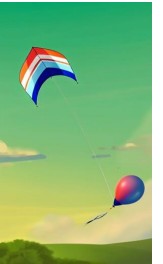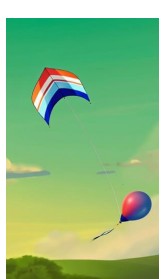

Figure 24: We scale up the test-time computation of Wan1.3B by $5\times$, ensuring equivalent inference times between Wan14B and **Wan1.3B+EvoSearch**. **EvoSearch** demonstrate superior text-alignment performance.

Prompt: A person's hair changes from black to blonde.

### *Wan14B*

### *Wan1.3B + EvoSearch*

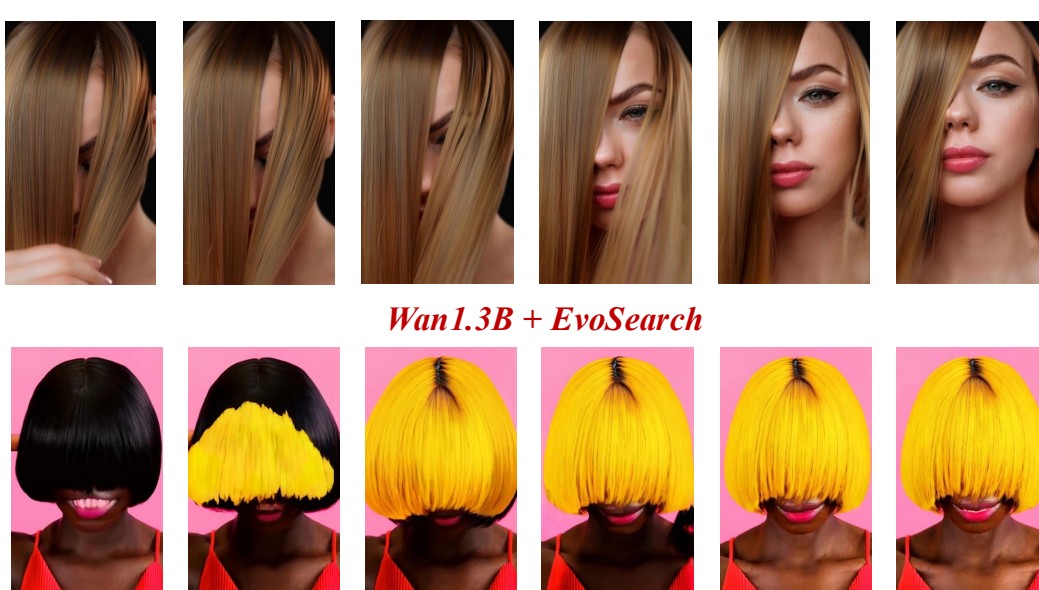

Figure 25: We scale up the test-time computation of Wan1.3B by $5\times$, ensuring equivalent inference times between Wan14B and **Wan1.3B+EvoSearch**. **EvoSearch** enhances Wan1.3B's capability in dynamic-attribute video generation.

Prompt: The plastic water cup turned into a metal water cup

### *Wan14B*

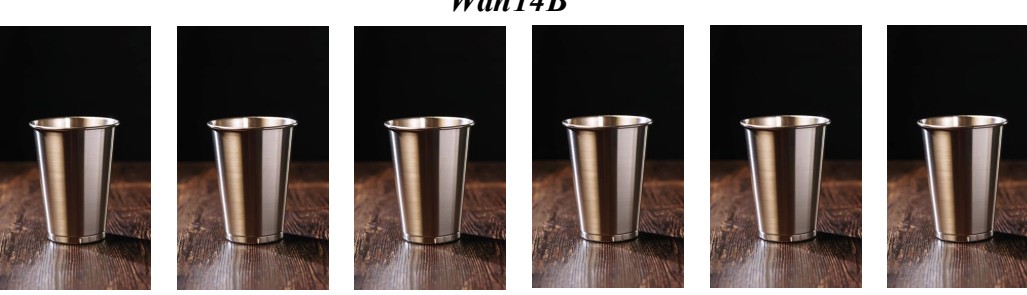

### *Wan1.3B + EvoSearch*

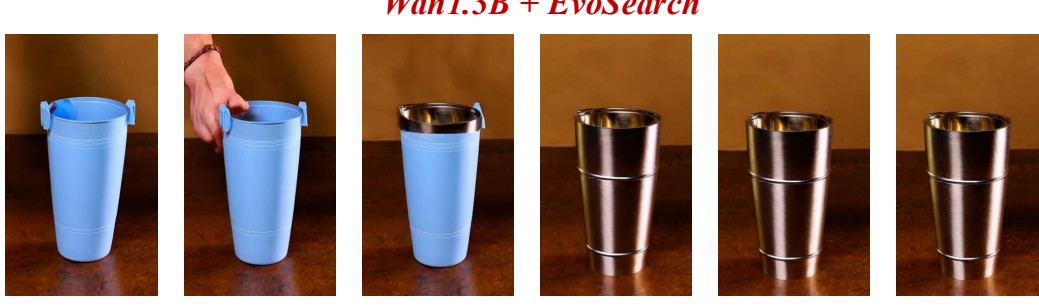

Figure 26: We scale up the test-time computation of Wan1.3B by $5\times$, ensuring equivalent inference times between Wan14B and **Wan1.3B+EvoSearch**. **EvoSearch** enhances Wan1.3B's capability in handling challenging prompts, outperforming Wan14B given the same inference time.

Prompt: A wooden toy is placed gently on the surface of a small bowl of water.

### *Wan14B*

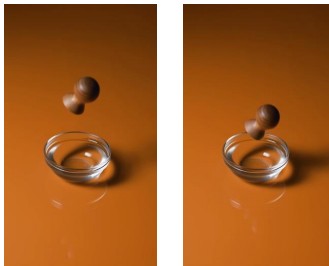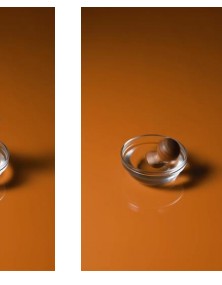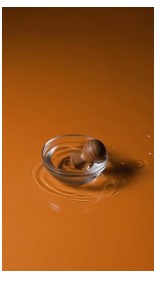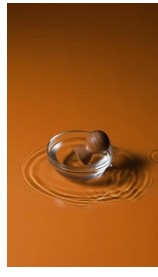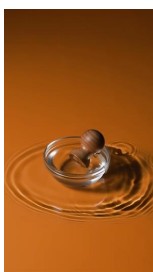

### *Wan1.3B + EvoSearch*

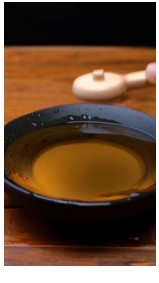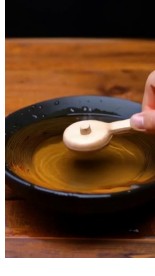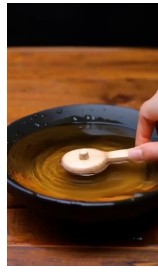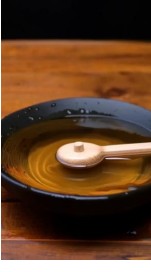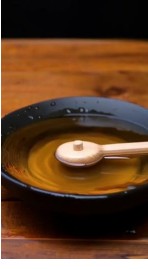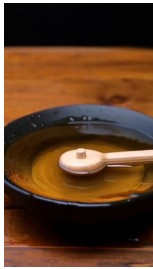

Figure 27: We scale up the test-time computation of Wan1.3B by $5\times$, ensuring equivalent inference times between Wan14B and **Wan1.3B+EvoSearch**. The video generated by **EvoSearch** follows the text instruction more closely, exhibiting improved logical consistency.

Prompt: A water droplet slides down the edge of a smooth sheet of aluminum, maintaining its spherical form

### *Wan14B*

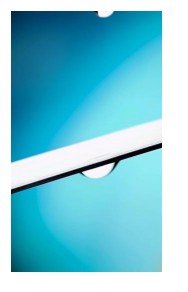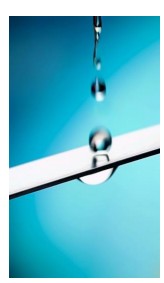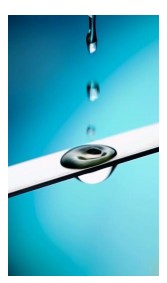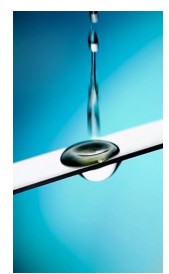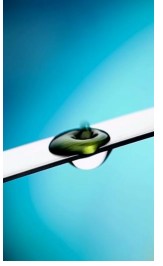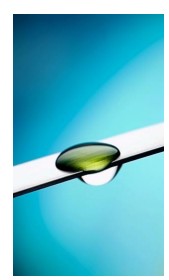

### *Wan1.3B + EvoSearch*

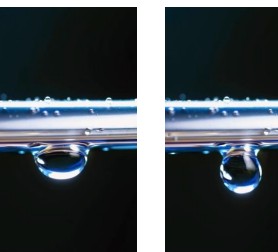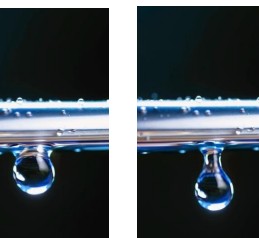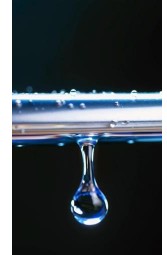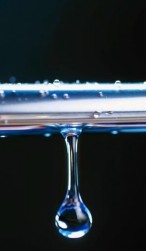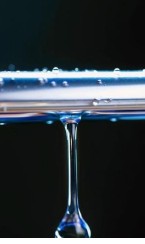

Figure 28: We scale up the test-time computation of Wan1.3B by $5\times$, ensuring equivalent inference times between Wan14B and **Wan1.3B+EvoSearch**. **EvoSearch** significantly improves the generation quality with superior semantic alignment.

