# OpenReview forum: "Scaling Image and Video Generation via Test-Time Evolutionary Search"
_ICLR.cc/2026/Conference — Submitted to ICLR 2026_

### Official Review · Reviewer_6YMr · 2025-10-23

**Soundness:** 2
**Presentation:** 2
**Contribution:** 1
**Rating:** 2
**Confidence:** 4

**Summary:**

The paper proposes EvoSearch, a method for improving image and video generation models through test-time evolutionary optimization. Rather than training or fine-tuning model weights, EvoSearch adapts the sampling process of diffusion models dynamically at inference. It formulates generation as a search problem where multiple candidate are evolved to maximize the reward. Empirical results on benchmarks show consistent gains in perceptual quality metrics.

**Strengths:**

The paper presents a training-free method to improve generative models at inference. EvoSearch applies to many diffusion architectures for both image and video generation and is among the first to study test-time scaling for video models, which broadens its impact across modalities. The experiments show consistent gains on perceptual quality metrics, indicating practical value.

**Weaknesses:**

While the proposed method brings additional performance gains across tasks, the method relies heavily on heuristic choices for the evolution schedule, mutation strategy, and initialization of noise parameters. How these parameters are chosen? Can authors give justifications for the parameters? Also, this may limit scalability for new model, task, and reward.

Although the method is new, test-time scaling has been explored extensively. The paper needs a more convincing case for why EvoSearch should work and why it should outperform existing methods such as best of N and particle sampling. The approach is framed as evolutionary optimization, but the paper does not explain in a principled way how the search improves the denoising trajectory. As written, the gains appear purely empirical. Providing more intuitions and formal derivations would strengthen the claims.

Lastly, running evolutionary search at test time introduces computational overhead, especially for video generation where reward and generative model are large. The paper lacks a thorough analysis of runtime and trade-offs compared to standard sampling methods.

Minor
- L.234, 304 Eq. equation typo

**Questions:**

Please see weaknesses.

---

> ### Author Response · Authors · 2025-11-20
> **Response to Reviewer 6YMr (Part 1)**
>
> We thank the reviewer for the time and effort made in the review! **We would like to clarify our contributions and address your concerns as follows**:
>
> >1. While the proposed method brings additional performance gains across tasks, the method relies heavily on heuristic choices for the evolution schedule, mutation strategy, and initialization of noise parameters. How these parameters are chosen? Can authors give justifications for the parameters? Also, this may limit scalability for new model, task, and reward.
>
> Thanks for your question. We respectfully disagree that our method relies heavily on heuristic choices. We would like to clarify the generalizability, reproducibility, and applicability of our method as follows:
> - **Evolution Schedule**: We include detailed ablation experiments for the evolution schedule in Appendix B.1 (Figures 12). Given the same NFEs, we observe that EvoSearch is highly insensitive to the specific choice of the evolution schedule. Different evolution schedule $\mathcal{T}$ only yields $1e^{-3}\sim1e^{-2}$ on the order of performance differences. Our empirical results demonstrate that an evolution schedule with uniform intervals yields relatively superior performance. Thus, we recommend a simple uniform interval schedule as a standard default. These ablation studies demonstrate that EvoSearch is robust across different hyper-parameter settings.
> - **Mutation Strategy**: Following our approach demonstrated in lines 314-323, we respectfully argue that our mutation strategy is not a heuristic choice, but a unified mechanism adaptable to any model.
>   -  **Initial Mutation ($\beta$)**: As shown in Eq.3, we set $\beta=0.3$ for the initial noise mutation. Crucially, this hyperparameter, $\beta$, is fixed across all experiments in the paper. The fact that a single value works for both Stable Diffusion and Flux (images) and Wan/Hunyuan (videos) demonstrates that this parameter does not limit scalability. We do not require any manual tuning for this parameter.
>   -  **Intermediate Mutation ($\sigma_t$)**: The mutation strength for intermediate steps is not a tunable hyperparameter. As detailed in lines 320-325, $\sigma_t$ is strictly aligned with the pre-defined noise schedule of the base model (SDE). This ensures that our mutation operator automatically adapts to the physics of any new model (e.g., SD2.1, Flux, Wan, etc.) without manual intervention. **Our proposed mutation strategies are theoretically validated, as discussed in Appendix A.2.**
> - **Initialization of Noise Parameters**: We sample initial noises from Gaussion distrbution $\mathcal{N}(0,I)$. `We do not employ any heuristic selection, "lucky seed" curation, or complex initialization tricks`. The diversity and quality of the results are driven entirely by the evolutionary search process (evaluation, selection, and mutation), not by engineered initialization.
>
> To directly address the concern about applying EvoSearch to new tasks:
> - We successfully applied EvoSearch to `4` different model families (SD2.1, Flux, Wan2.1, Hunyuan) and `2` modalities (Image, Video) using `8` different reward functions (ImageReward, ClipScore, Aesthetic, HPSv2, GenEval, DPGBench, VBench, VBench2).
> - Key Point: All these diverse experiments used the exact same configuration (Uniform evolution schedule, $\beta=0.3$).
>
> Overall, this empirical evidence confirms that EvoSearch does not require per-task hyperparameter tuning, making it highly scalable and easy for future researchers to reproduce. Additionally, `we will open-source all the code upon publication`.

---

> > ### Author Response · Authors · 2025-11-20
> > **Response to Reviewer 6YMr (Part 2)**
> >
> > >2. The paper needs a more convincing case for why EvoSearch should work and why it should outperform existing methods such as best of N and particle sampling ... Providing more intuitions and formal derivations would strengthen the claims.
> >
> > Thanks for your valuable suggestion. We would like to clarify that EvoSearch outperforms Best-of-N and Particle Sampling due to three fundamental advantages: **Active Exploration Based on Evolutionary Process**, **Reward Fidelity**, and **Computation Reuse**.
> >
> > **1. Overcoming Particle Impoverishment via Active Exploration (vs. Particle Sampling, Best-of-N)**
> >
> > - **The Limitation of Baselines**: Standard Particle Sampling (e.g., FK-Steering) relies on resampling high-weight particles. This leads to a well-known issue in Sequential Monte Carlo methods called "particle impoverishment" or "collapse," where the population degenerates into multiple identical copies of a few "good" particles. This restricts the search to the initial support of the distribution. **Best-of-$N$** is a "blind" brute-force approach. It generates $N$ independent trajectories from start to finish ($T\to 0$) without integrating any optimization process. It wastes massive amounts of computation fully generating low-quality samples that are ultimately discarded.
> > - **The EvoSearch Advantage**: EvoSearch solves this by introducing Mutation (derived from the SDE diffusion term). Instead of simply duplicating a good parent, we perturb it to explore its local neighborhood. By reinterpreting the denoising trajectory as an evolutionary path, both the initial noise $x_T$ and the intermediate state $x_t$ can be evolved towards higher-quality generation.
> > - Intuition (Fig. 4): As shown in Figure 4, high-reward solutions in the latent space tend to be clustered (i.e., if $x$ is good, then the states in its neighborhood are likely better). EvoSearch actively exploits these "basins of attraction" via mutation, enabling it to discover superior states that lie outside the original particle pool—something standard resampling cannot do.
> >
> > **2. Eliminating Reward Estimation error**
> >
> > - **The Limitation of Baselines** : Particle Sampling requires assigning weights at intermediate steps $t$. Since the final image $x_0$ is unknown, these methods rely on noisy approximations (e.g., Tweedie’s formula $\mathbb{E}[x_0|x_t]$) for reward estimation. These approximations are highly inaccurate in early steps, leading to "cumulative errors", where the particle sampling algorithm commits to suboptimal denoising paths based on false reward signals.
> > - **The EvoSearch Advantage**: EvoSearch evaluates the fully denoised $x_0$. This ensures 100% reward signal fidelity. Our selection decisions are based on rewards calculated on clean images, eliminating the bias and variance inherent in the approximations used by particle sampling.
> >
> > **3. Computational Efficiency**
> >
> > - **The Limitation of Baselines**: Baselines, including Best-of-N and particle sampling, are a passive filtering approach, which are computationally wasteful, as they expend a large amount of computation generating complete trajectories for samples that ultimately yield low rewards.
> > - **The EvoSearch Advantage**: We implement a unique Caching Mechanism (detailed in Algorithm 2). EvoSearch caches intermediate states at pre-defined evolutionary timesteps. When a "parent" is selected, we can resume from the cached states to expand the parent population size and start from the intermediate step $t$. This effectively allows us to "reuse" the denoised states of previous generations while only spending compute on remaining denoising steps.
> >
> > **Summary**
> >
> > Overall, our work is the first to formulate the test-time scaling problem for both image and video generation as an evolutionary search problem. EvoSearch makes distinctive contributions by innovatively introducing evolutionary concepts (e.g., mutation, selection, and evaluation) into denoising problems. **By replacing passive filtering (resampling) with active search (mutation) and replacing noisy approximations with trustworthy evaluation, EvoSearch provides a theoretically more robust framework for maximizing generation quality**.

---

> ### Author Response · Authors · 2025-11-20
> **Response to Reviewer 6YMr (Part 3)**
>
> >3. ... evolutionary search at test time introduces computational overhead, especially for video generation where reward and generative model are large. The paper lacks a thorough analysis of runtime and trade-offs compared to standard sampling methods.
>
> We appreciate the reviewer’s focus on the practical trade-offs, particularly for computationally intensive tasks like video generation. We address the concerns regarding runtime and efficiency from two perspectives: the research context and the empirical efficiency.
>
> **1. Context: Test-Time Scaling Paradigm**
>
> We would like to clarify that our paper focuses on **advancing the field of test-time scaling in visual generation** (including both image and video). Similar to how training-time scaling laws utilize massive compute to improve base models, test-time scaling investigates how much generation performance can be improved by trading off computation for quality during inference. Therefore, **requiring additional computational budget is an inherent characteristic of this research direction, not a limitation**. Our goal is to determine the upper bounds of generation quality when compute is not the bottleneck (e.g., for commercial video production or high-end asset production), rather than merely developing faster inference/sampling methods. We hope this aspect can be evaluated objectively.
>
> **2. Empirical Efficiency**
>
> Contrary to the concern that the overhead makes the method impractical, our analysis in Appendix B.2 demonstrates that EvoSearch is highly efficient when improving quality. We observe that EvoSearch achieves superior performance compared with baselines given the same wall-clock time, demonstrating the efficiency and effectiveness of our proposed method. As shown in our experiments, standard baselines like Particle Sampling and Best-of-N rapidly hit a performance plateau. In contrast, EvoSearch continues to improve generation quality as the computational budget increases. This makes it a more "resource-efficient" scaler: every unit of extra compute yields a tangible quality return.
>
> **3. The Video Generation Trade-off**
>
> Specifically regarding video generation, we demonstrated a critical trade-off in our experiments: EvoSearch allows a small model (Wan-1.3B) to outperform a much larger model (Wan-14B), as shown in lines 487-497 in our paper. Running EvoSearch on a 1.3B model is often computationally cheaper (and has lower VRAM requirements) than running a single inference pass on a 14B model, yet it achieves superior results. This proves that EvoSearch offers a practical path to state-of-the-art video quality without requiring massive proprietary models.
>
>
> To further demonstrate the computation efficiency of our method, we added additional experiments to compare the required wall-clock time and NFEs of different methods to reach the same ImageReward (i.e., 1.46) on DrawBench. We record the wall-clock time on the same machine to ensure fairness. **The following experiments demonstrate that EvoSearch costs less time than all the baselines, showing the superior computation efficiency of our method.** The efficiency of EvoSearch stems from its streamlined fitness evaluation mechanism: EvoSearch requires only one reward computation per sample (at the end of the denoising generation). In contrast, RBF and particle sampling methods demand multiple reward computations and resampling operations per sample at multiple intermediate steps, resulting in significantly higher computational overhead.
>
> | Methods|Required NFEs|Required Wall-Clock Time |ImageReward on DrawBench|
> | -| -| -|-|
> | EvoSearch|	**$50\times 90$**|	**85.6s**    |1.46|
> | Particle Sampling|	$50\times 100$|	117.7s    |1.46|
> | Best of N|	$50\times 1200$|	971.4s    |1.46|
>
> >4. L.234, 304 Eq. equation typo
>
> Thanks for your comment! We have fixed the typo in the updated version of our paper.
>
> ---
>
>
> **Overall, we hope that our responses above have addressed your concerns. If they have, we would appreciate it if you could kindly consider raising the score. If not, we are always willing to address any of your further concerns. Thank you again for reviewing our work!**

---

> ### Author Response · Authors · 2025-11-26
> **Looking forward to your reply!**
>
> Dear Reviewer 6YMr,
>
> As the discussion period will end soon and there have been over 5 days since we posted our rebuttal, we'd like to ask if we have addressed your concerns. We believe that we have clarified the concerns and questions raised.
>
> Specifically, we provided detailed justifications for the parameters used in our paper, showing our method is robust across different settings. We provided comprehensive discussions to demonstrate the design principle and advantages of our method compared with baselines. We also add new experiments to validate the superior computational efficiency of our method compared with baselines.
>
> If you have any additional thoughts or questions about our responses, we would be more than happy to address them.
>
> Thank you once again for your insightful review! We look forward to your further feedback.
>
> Best,
>
> The authors

---

### Official Review · Reviewer_KnUy · 2025-10-29

**Soundness:** 3
**Presentation:** 2
**Contribution:** 2
**Rating:** 6
**Confidence:** 3

**Summary:**

The paper proposes a new type of test-time scaling (TTS) method for diffusion models, inspired by evolutionary algorithms. Unlike previous approaches that require $N$-times sampling (Best-of-$N$) or are restricted to initial ODE stages (particle sampling), the proposed method enables multiple sampling trajectories starting from intermediate states $x_t$, with the ability to modify the initial noise based on intermediate reward evaluations. For both image and video generation with flow-based models (noting that denoising diffusion models are a special case of flow-based models), the proposed method achieves superior quality and diversity.

**Strengths:**

- Performance is promising.
- The paper provides extensive analysis including toy example and ablation study with hyper-parameters.
- The proposed idea seems to be novel.

**Weaknesses:**

- The writing could be improved. Although the paper focuses on reinterpreting TTS from the perspective of evolutionary algorithms, the terminology used is unconventional. Consequently, the roles of hyperparameters such as the population scheduler $K$ and evolution schedule $T$ are unclear and difficult to interpret.

- Line 265: Chung et al., 2023 should be corrected to [1].

- Computing the reward on the fully denoised $x_0$ (lines 264–268) is not new. [2], which is missing from the related work, also performs evaluation on the fully denoised $x_0$.

- Lines 192–194: the paper claims that $N$-times sampling for Best-of-$N$ is computationally inefficient. However, the proposed method also performs repeated sampling for evaluation (lines 259–268). While this may be slightly more efficient since sampling starts from an intermediate step $t$ instead of the initial state $x_T$, the main source of computational overhead remains comparable to previous methods.

- The paper suggests that using an SDE is necessary for exploration. However, this has already been explored in [3], which demonstrated that SDE-based dynamics are required for fine-tuning diffusion models.

Reference

[1] Noise2Score: Tweedie's Approach to Self-Supervised Image Denoising without Clean Images, NeurIPS 2021

[2] Training Diffusion Models Towards Diverse Image Generation with Reinforcement Learning, CVPR 2024

[3] Adjoint Matching: Fine-tuning Flow and Diffusion Generative Models with Memoryless Stochastic Optimal Control, ICLR 2025

**Questions:**

- Could the authors provide guidance on how to select appropriate hyperparameters when applying EvoSearch to a new generative model?

---

> ### Author Response · Authors · 2025-11-20
> **Response to Reviewer KnUy (Part 1)**
>
> Thanks for your positive assessment of our work, and for noting that our work is novel and the results are promising! We carefully address your concerns as follows:
>
> >1. The writing could be improved. Although the paper focuses on reinterpreting TTS from the perspective of evolutionary algorithms, the terminology used is unconventional. Consequently, the roles of hyperparameters such as the population scheduler $K$ and evolution schedule $T$ are unclear and difficult to interpret.
>
> Thanks for your suggestion. We will improve the writing in the next version of our paper.
>
> The population schedule $\mathcal{K}$ controls the width of the search space throughout the evolutionary generation process. Specifically, it determines the population size (number of parent particles) maintained at each evolution timestep.
>
> The evolution schedule $\mathcal{T}$ controls the timing of evolutionary search. It specifies exactly which denoising timesteps undergo the evolutionary search (evaluation, selection, and mutation) versus standard denoising. This allows us to concentrate computational effort on the most critical steps of the generation trajectory (e.g., early steps for overall structure generation).
>
> The flexible design of these two schedules is central to EvoSearch’s adaptability across different tasks. Unlike rigid baselines, EvoSearch allows users to dynamically allocate their computation budget—spending more compute (higher population size) on critical steps (specific $T$) to maximize quality under varying constraints.
>
> >2. Line 265: Chung et al., 2023 should be corrected to [1].
>
> Thanks for your valuable suggestion. We have updated the citation in the revised manuscript.
>
> >3. Computing the reward on the fully denoised $x_0$
>  (lines 264–268) is not new. [2], which is missing from the related work, also performs evaluation on the fully denoised $x_0$.
>
> Thanks for your comment. We have updated the related work to include this work for discussion. While computing the reward on $x_0$ is not new in broader literature, our contribution lies in how this is applied to **test-time scaling** to overcome the limitations of previous methods in this field.
>
> Existing particle filtering methods (e.g., FK-Steering) typically rely on approximated $x_0$ (often derived via one-step estimation/Tweedie’s formula from $x_t$) to calculate rewards during the intermediate steps. However, these intermediate approximations are often noisy and inaccurate, particularly in the early stages of generation, leading to sub-optimal selection decisions. EvoSearch eliminates this estimation error without compromising computation efficiency due to our carefully designed evolution mechanisms.

---

> > ### Author Response · Authors · 2025-11-20
> > **Response to Reviewer KnUy (Part 2)**
> >
> > >4. Lines 192–194: the paper claims that $N$-times sampling for Best-of-$N$
> >  is computationally inefficient. However, the proposed method also performs repeated sampling for evaluation (lines 259–268). While this may be slightly more efficient since sampling starts from an intermediate step $t$ instead of the initial state $x_T$, the main source of computational overhead remains comparable to previous methods.
> >
> > Thanks for your insightful comment. The main computation budget lies in the denoising (sampling) steps. This is general for all test-time scaling methods for denoising models. However, we argue that EvoSearch is more computationally efficient (not comparable) than previous methods.
> >
> > **Best-of-$N$** is a "blind" brute-force approach. It generates $N$ independent trajectories from start to finish ($T\to 0$) without integrating any optimization process. It wastes massive amounts of computation fully generating low-quality samples that are ultimately discarded.
> >
> > **EvoSearch** is a "guided" evolutionary process. By evaluating and selecting parents at intermediate steps $t$, we actively prune unpromising trajectories early. This ensures that the computational budget is concentrated solely on the most promising paths (Elites) and their mutated offspring.
> >
> > The reviewer notes that starting from intermediate step $t$ is "slightly" more efficient. We argue this difference is substantial. If an evolutionary step occurs at $t=20$ (in a 50-step denoising process), obtaining the final samples costs only 60% of the compute required for a full Best-of-$N$ generation. This allows us to explore a much larger population of candidates for the same computational cost.
> >
> > To further demonstrate the computation efficiency of our method, we added additional experiments to compare the required wall-clock time and NFEs of different methods to reach the same ImageReward (i.e., 1.46) on DrawBench. We record the wall-clock time on the same machine to ensure fairness. **The following experiments demonstrate that EvoSearch costs less time than all the baselines, showing the superior computation efficiency of our method**. The efficiency of EvoSearch stems from its streamlined fitness evaluation mechanism: EvoSearch requires only one reward computation per sample (at the end of the denoising generation). In contrast, RBF and particle sampling methods demand multiple reward computations and resampling operations per sample at multiple intermediate steps, resulting in significantly higher computational overhead.
> >
> > | Methods|Required NFEs|Required Wall-Clock Time |ImageReward on DrawBench|
> > | -| -| -|-|
> > | EvoSearch|	**$50\times 90$**|	**85.6s**    |1.46|
> > | Particle Sampling|	$50\times 100$|	117.7s    |1.46|
> > | Best of N|	$50\times 1200$|	971.4s    |1.46|
> >
> > >5. The paper suggests that using an SDE is necessary for exploration. However, this has already been explored in [3], which demonstrated that SDE-based dynamics are required for fine-tuning diffusion models.
> >
> > We thank the reviewer for pointing out this relevant work. We have already cited this work for noting it as a stochastic optimization work. We would like to clarify the distinction between the findings in this work and our contribution.
> > - This work demonstrates that SDE dynamics are critical for fine-tuning diffusion models. However, its results are based on parameter tuning, including sampling, loss computation, and gradient backpropagation.
> > - In contrast, our work focuses on the test-time inference phase without parameter updates. We uniquely leverage the SDE-based dynamic not merely for enhanced generation diversity, but specifically for the designing of the mutation operator within an evolutionary framework. This allows us to leverage SDEs to actively explore the search space for reward maximization, a distinct application from the fine-tuning context explored in previous works.

---

> > > ### Author Response · Authors · 2025-11-20
> > > **Response to Reviewer KnUy (Part 3)**
> > >
> > > >6. Could the authors provide guidance on how to select appropriate hyperparameters when applying EvoSearch to a new generative model?
> > >
> > > Thanks for your question. Based on our extensive ablation studies in Appendix B.1, we find that EvoSearch is remarkably robust to hyperparameter variations, which simplifies its application to new generative models. We offer the following concrete guidelines for implementation:
> > > - For population size schedule $\mathcal{K}$, we recommend a "front-heavy" strategy. Set the population size in the first generation of the evolutionary search to be $2\times$ larger than in subsequent generations (e.g., starting with 20 candidates, and 10 for remaining steps). Our experiments reveal that maximizing diversity in the initial phase is critical. Once a high-quality trajectory is established, a smaller population is sufficient to refine it.
> > > - For evolution schedule $\mathcal{T}$, we recommend a uniform interval schedule. Our empirical results demonstrate that the method is relatively insensitive to the exact timing of the evolution steps. A simple uniform distribution yields superior performance without the need for manual tuning.
> > >
> > > Our ablation studies confirm that EvoSearch does not require complex, model-specific hyperparameter search. Future research can adopt these recommended configurations as a standard baseline. Additionally, `we will open-source all the code upon publication to ensure reproducibility`.
> > >
> > > ---
> > >
> > > **We hope to have addressed all the raised concerns and would be happy to respond to further questions and suggestions. Thank you again for your time and efforts, and for your support of our work!**

---

> ### Author Response · Authors · 2025-11-26
> **Looking forward to your reply!**
>
> Dear Reviewer KnUy,
>
> We hope this message finds you well. We are writing to kindly follow up on our response to your valuable comments and questions, which we submitted over 5 days ago.
>
> In our rebuttal, we provided detailed explanations of our method, updated the related work section in our paper, and added new experiments to validate the computational efficiency of our method compared with baselines.
>
> Your feedback has been crucial in refining our work, and we greatly appreciate the time and effort you have invested in reviewing our paper. If you have any additional thoughts or questions about our responses, we would be more than happy to address them.
>
> Thank you once again for your insightful review! We look forward to your further feedback.
>
> Best,
>
> The authors

---

### Official Review · Reviewer_oKyW · 2025-10-30

**Soundness:** 3
**Presentation:** 2
**Contribution:** 3
**Rating:** 6
**Confidence:** 2

**Summary:**

This paper explores test-time scaling (TTS) for vision generative models and proposes EvoSearch, an evolutionary search–based approach for enhancing the denoising trajectory in both diffusion and flow-based generative models. Instead of modifying model architecture or training objective, EvoSearch leverages evolutionary mechanisms during inference to identify better intermediate latent states as parents for subsequent denoising steps. The paper presents extensive experiments across image and video generation tasks, showing improvements in visual quality and sample diversity.

**Strengths:**

- Novelty: Using evolutionary algorithms to optimize denoising trajectories during inference is original and conceptually appealing.
- Strong empirical evaluation: Extensive experiments, ablations, and analysis support the method's effectiveness.
- General applicability: Works across diffusion and flow-based models, and both images and videos.
- No training overhead: Fits well within the TTS paradigm—improving sampling quality without retraining.

**Weaknesses:**

- Clarity of algorithm description: The current presentation of EvoSearch's workflow, particularly in the overview figure and pseudocode, lacks clarity and makes it difficult to precisely understand the evolutionary operators.
- Ambiguous visualization (Figure 3): The diagram includes unexplained symbols (e.g., check marks, arrows, population filtering meaning, mutation signs), making the process non-transparent. It would strongly benefit from a redesign with clearer semantics and annotations explaining the denoising vs. generation stages.
- Reward computation description unclear: There is confusion regarding whether the reward r(x_0) is computed per-parent or aggregated, and how it aligns with the expected reward definition under p(x_0|x_t). This should be explicitly clarified step-by-step.

**Questions:**

- What does the check mark on the middle image mean?
- Why is the reward value in selection shown as 0.01, and why are 3 elites highlighted?
- In the mutation block, what do the different operators (→, =, + etc.) represent?

Reward Computation in Algorithm 2:
At t = 0, if we have 10 parent states, do we compute 10 rewards or 1 aggregated reward?
- If only 1 reward is computed, what does it correspond to?
- If 10 rewards are computed, how does this align with Eq.2's expectation over p(x_0, x_t)?

---

> ### Author Response · Authors · 2025-11-20
> **Response to Reviewer oKyW (Part 1)**
>
> We thank the reviewer for the positive assessment and for noting the novelty, strong empirical results, and generalizability of our work! Here, we carefully address your concerns below.
> >1. Clarity of algorithm description: The current presentation of EvoSearch's workflow, particularly in the overview figure and pseudocode, lacks clarity and makes it difficult to precisely understand the evolutionary operators.
>
> Thanks for your question. We have made several updates to the overview figure and pseudocode to improve the clarity, mainly by adding annotations.
>
> Unlike previous methods that required separate test-time scaling strategies and manual tuning for diffusion and flow-based models, EvoSearch is a unified approach that works seamlessly across different model architectures and generalizes well to different modalities. EvoSearch contains three key operators: (1) **evaluation operations**. At evolution timestep $t_i$, EvoSearch performs denoising from $t_i\to 0$ for fitness evaluation on clean samples for obtaining reliable and trustworthy rewards. (2) **Tournament selection** strategy that balances quality and diversity, which has been proven effective in previous literature in evolutionary algorithms [1,2]. (3) **Mutation operations** include separate designs for initial noises and intermediate denoising states. The proposed mutation mechanisms aim to create totally new samples of high quality across each evolution branch.
>
> **Reference**
>
> [1] Miao et al., Evolving Reinforcement Learning Algorithms, ICLR 2021
>
> [2] David E Goldberg and Kalyanmoy Deb. A comparative analysis of selection schemes used in genetic algorithms. FOGA, 1991.
> >2. Ambiguous visualization (Figure 3): The diagram includes unexplained symbols (e.g., check marks, arrows, population filtering meaning, and mutation signs), making the process non-transparent. It would strongly benefit from a redesign with clearer semantics and annotations explaining the denoising vs. generation stages.
>
> Thanks for your question. We have updated the figure by adding annotations for these symbols to improve clarity. We also updated the caption for a better grasp of our method. We provide brief explanations for the symbols you mentioned as follows:
> - The check marks mean that the generated images are selected for the best quality.
> - The arrows in the figure denote the denoising process. And the arrows in the 'evosearch' block just indicate the progressive relationship.
> - $x_t^{\rm parent}$ denote the parent population before evolutionary generation, while $x_t^{\rm child}$ is the evolutionaized new denoising states.
> - For the signs in the mutation block, the green arrow means that the selected elite is preserved without any mutation. '+' and '=' means mutation operations defined in Eq.3 and Eq.4 in our paper.
>
> >3. Reward computation description unclear: There is confusion regarding whether the reward r(x_0) is computed per-parent or aggregated, and how it aligns with the expected reward definition under p(x_0|x_t). This should be explicitly clarified step-by-step.
>
> Thanks for your question. We clarify that the reward $r(x_0)$ is computed per-parent (per single sample). Mathematically, the ideal objective is to maximize the expected reward over the distribution of possible clean images $x_0$ given the current noisy state $x_{t}$, i.e., $\\mathbb{E}_{{x}_0 \sim p_0(x_0|x_t)} [r(x_0) | x_t]$. However, computing the exact expectation is computationally prohibitive, as it would require generating multiple full trajectories for every candidate at each intermediate denoising step. Therefore, in our implementation, we use a single-sample Monte Carlo approximation. We estimate the expectation using a single sample $x_0$. To be explicit, our evaluation process operates as follows:
> - **Step 1** (Generation): For a specific parent (noisy state $x_t$), we complete the denoising process to generate a single clean image $x_0$.
> - **Step 2** (Evaluation): We calculate the reward $r(x_0)$ for the denoised image (e.g., its CLIP score or ImageReward).
> - **Step 3** (Selection): This scalar value $r(x_0)$ serves as the fitness score for that parent for tournament selection. Although this is a stochastic estimate of the true expectation, the evolutionary population dynamics allow the algorithm to robustly optimize the objective despite the variance in individual estimates.
>
> We have updated the text surrounding Eq. (2) and the Algorithm to further improve the clarity.

---

> > ### Author Response · Authors · 2025-11-20
> > **Response to Reviewer oKyW (Part 2)**
> >
> > >4. What does the check mark on the middle image mean?
> >
> > Thanks for your question. We apologize for the lack of explicit clarification in the figure caption. The check mark indicates the final selected image resulting from the complete evolutionary search process. The image is selected since it exhibits higher quality than that of earlier generations, e.g., generation 1 or 2. This visualization is designed to demonstrate the progressive improvement of EvoSearch. By contrasting the standard denoising process, we highlight how the generation quality consistently improves with the progression of the evolutionary search.
> >
> > We have updated the figure caption to explicitly state this definition to avoid future confusion.
> >
> > >5. Why is the reward value in selection shown as 0.01, and why are 3 elites highlighted?
> >
> > Thanks for your question. The reward value 0.01 is given by ImageReward, which reflects the human preference given the prompt and generated image. Because this score is low, this candidate (**labeled as Image #1**) is rejected during the selection process. It is included in the figure solely to illustrate a discarded sample in contrast to the selected ones. **Image #3** is highlighted because it achieves the highest reward score of 0.68, qualifying it as an elite, which is retained for the next generation.
> >
> > In summary, the figure demonstrates the selection logic: Image #3 (Score 0.68) is selected as an elite, while Image #1 (Score 0.01) is discarded. Note that Image #3 is not equal to the image with reward 0.01. The image with reward 0.01 is Image #1.
> >
> >
> > >6. In the mutation block, what do the different operators (→, =, + etc.) represent?
> >
> > Thanks for your question. → means we preserve the parent without mutation (typically applied to the selected "elites" to preserve the best-performing candidate). + and = represent the specific Mutation operations defined in our method. These operators correspond to the mathematical formulations in Eq. (3) and Eq. (4), where noise is added or modified to generate novel offspring. The goal of these operations is to introduce exploration, allowing the algorithm to expand the latent space for higher-quality solutions.
> >
> > >7. Reward Computation in Algorithm 2: At t = 0, if we have 10 parent states, do we compute 10 rewards or 1 aggregated reward?
> >
> > Thanks for your question. We clarify that we compute distinct rewards for each parent individually. Therefore, if there are 10 parent states, we calculate 10 separate reward values. The reward function is applied to the decoded image of each candidate independently to determine its specific fitness for the selection process. No aggregation across parents is performed.
> >
> > ---
> >
> > We have updated our paper based on your valuable feedback. **We hope to have addressed all the raised concerns and would be happy to respond to further questions and suggestions. Thank you again for your time and efforts, and for your support of our work!**

---

> ### Author Response · Authors · 2025-11-26
> **Looking forward to your reply!**
>
> Dear Reviewer oKyW,
>
> We hope this message finds you well. We are writing to kindly follow up on our response to your valuable comments and questions, which we submitted over 5 days ago.
>
> In our rebuttal, we updated the visualization in our paper by adding more annotations, and providing detailed illustrations to demonstrate our method clearly.
>
> Your feedback has been crucial in refining our work, and we greatly appreciate the time and effort you have invested in reviewing our paper. If you have any additional thoughts or questions about our responses, we would be more than happy to address them.
>
> Thank you once again for your insightful review! We look forward to your further feedback.
>
> Best,
>
> The authors

---

### Official Review · Reviewer_XkkV · 2025-11-01

**Soundness:** 2
**Presentation:** 3
**Contribution:** 2
**Rating:** 4
**Confidence:** 3

**Summary:**

This work proposes Evolutionary Search (EvoSearch), a novel test-time scaling framework for generative models in images and videos. Instead of training new models or fine-tuning, EvoSearch allocates extra computation at inference to improve output quality by iteratively refining generations. It treats the diffusion/flow model’s denoising process as an evolutionary algorithm: a population of sample trajectories is evolved through selection and mutation at each step. This approach aims to preserve diversity while gradually steering samples toward higher reward. EvoSearch is a general method applicable to both diffusion models (e.g. SD 2.1) and flow-based generative models (e.g. Flux) for image and video generation. Empirically, the authors show that as inference compute (“number of function evaluations”, NFEs) increases, EvoSearch produces higher-quality outputs that outperform baseline strategies like best-of-$N$ and particle filtering on several benchmarks. Notably, using EvoSearch, a smaller text-to-video model (Wan 1.3B) was able to match or surpass the performance of a 10× larger model (Wan 14B) given the same inference-time budget. The paper positions EvoSearch as a general, training-free approach to test-time alignment and improved generation quality for vision models, drawing inspiration from biological evolution to navigate the high-dimensional search space of generative model outputs.

**Strengths:**

EvoSearch introduces a fresh perspective by framing inference-time optimization as an evolutionary process. It leverages selection and mutation operators tailored to the diffusion denoising trajectory, which is an original way to maintain diversity and avoid the collapse seen in earlier methods.

When enough computation is allocated, EvoSearch does improve generation quality. The experiments show notable gains in output metrics as the number of inference steps increases. The experimental results indicate that given a high test-time compute budget, EvoSearch can find higher-reward samples than prior methods, validating its effectiveness in principle.

The evolutionary strategy explicitly tries to avoid the diversity collapse problem. By always selecting a population of top samples and mutating them with randomness, EvoSearch preserves multiple promising candidates instead of focusing on a single trajectory. The authors note that previous particle methods were limited by a fixed initial pool and could converge to similar results. In contrast, EvoSearch’s design (inspired by natural evolution) inherently maintains population diversity, which is important for generative tasks to not over-optimize the reward at the cost of mode collapse.

**Weaknesses:**

While EvoSearch is novel in framing the problem as an evolutionary algorithm, the practical advantages over existing particle sampling methods are not very convincing. In the Stable Diffusion 2.1 image experiments, EvoSearch’s improvement over a standard particle filtering baseline is quite small. One could argue EvoSearch is a variant of particle sampling with an evolutionary selection twist, yielding comparable results to known methods on images. This weakens the claim of a significant contribution. Also, in FLUX experiments, EvoSearch’s gains come at the cost of very high computation, raising concerns about practicality. The authors had to go up to around 5000 NFEs before EvoSearch clearly overtook the particle sampling baseline, which is an enormous number. In terms of wall-clock time, this is on the order of 20+ minutes per single image generation (thousands of seconds on a single GPU) for one prompt. This dramatically limits real-world usability.

Also, it formulates the guidance as a reward (from models like ImageReward or CLIP score) and uses a gradient-free evolutionary search to optimize it. However, many of these reward models are differentiable, which raises the question: why not use gradient-based optimization? Recent works have shown that directly optimizing the initial noise or latent via gradient ascent on a differentiable objective can be extremely effective. For instance, ReNO (Eyring et al., 2024) optimizes the initial noise in a one-step diffusion model by backpropagating the reward gradient, achieving significant gains in prompt fidelity within just ~50 iterations (around 20–50 seconds of compute). Likewise, D-Flow (Ben-Hamu et al., 2024) differentiates through the entire generative process of a diffusion/flow model to adjust the noise, effectively guiding generation by continuous optimization rather than random mutation. These approaches show that gradient-based test-time alignment is feasible and often faster or more direct than heuristic search. The fact that EvoSearch does not compare against any such differentiable methods is a notable omission. It’s unclear if the authors avoided gradients due to concerns like non-convexity or local optima; regardless, no discussion is provided. This is a weakness because the paper doesn’t justify its design choice of an evolutionary (gradient-free) strategy when, in principle, the reward signal is differentiable. Without comparing to methods like ReNO or D-Flow, we don’t know if EvoSearch’s gradient-free approach is actually more effective, or if it might be outperformed by simply taking gradient steps on the noise (which could potentially reach a high-reward solution much faster).

The authors did not include a comparison with the latest particle sampling advancement, namely the Rollover Budget Forcing (RBF) method introduced by Kim et al. (2025). RBF is specifically designed to improve inference-time sampling by adaptively reallocating the diffusion steps budget, and it has been shown to outperform prior particle filtering approaches. In fact, the paper’s related work section cites this method (Kim et al., 2025b) and notes that it “demonstrates superior results” over naive baselines. By failing to compare against RBF in the experiments, the evaluation is incomplete, and we don’t see how EvoSearch stacks up against the state-of-the-art in test-time sampling. Given that RBF was published in March 2025 and targets a very similar problem (scaling diffusion/flow model inference), it should have been included as a baseline. This omission is critical because RBF could potentially already solve some challenges (e.g., better compute allocation, maintaining diversity) that EvoSearch addresses.

The paper aspires to present a general-purpose test-time scaling approach, but in doing so it may be spreading its contributions thin. The most compelling results are in the video generation domain, where test-time optimization isnovel. However, for image generation, the method doesn’t substantially advance beyond what was already known (as discussed above). This disparity suggests that the paper’s contribution might have been better framed with a narrower scope: for example, focusing on test-time evolutionary search for video generation as the primary innovation.

**Questions:**

Have you considered leveraging gradient-based optimization on the reward function instead of (or in addition to) evolutionary search?

The results indicate EvoSearch can require thousands of steps (and very long wall-clock time) to fully exploit the reward. How practical is this method, and are there ways to reduce the compute required?

---

> ### Author Response · Authors · 2025-11-20
> **Response to Reviewer XkkV (Part 1)**
>
> We thank the reviewer for the valuable feedback and suggestions! We appreciate the reviewer for noting our work is novel, effective, and mitigates mode collapse. We carefully address your concerns below.
> >1. ...the practical advantages over existing particle sampling methods are not very convincing...
>
> Thanks for your comment regarding the magnitude of our improvements compared with the particle sampling baseline. We clarify our contributions in empirical performance as follows:
> - **Significance of Improvements over the Baseline**: First, it is important to clarify that our particle sampling baseline is implemented based on **FK-Steering** [1] (stated in lines 373-374), which is currently a strong, state-of-the-art method in this domain. Consequently, consistently outperforming such a robust baseline across diverse tasks is a non-trivial achievement for the community of test-time scaling. Contrary to the impression that the gains are "small," we argue that our empirical results demonstrate statistically significant margins on Stable Diffusion 2.1 image experiments:
>    - As shown in Fig. 5, EvoSearch outperforms particle sampling by `9.6\%` given $1e^4$ NFEs using ClipScore as the guidance (0.34 vs. 0.31). This gap widens to `10.7\%` (0.362 vs. 0.327) at $6e^4$ NFEs. Even on ImageReward, we maintain a consistent advantage of 3.2\% at $1e^4$ NFEs (1.61 vs. 1.56).
>    - We remark that EvoSearch consistently outperforms all the baselines as computation increases across different guidance rewards, showing its robustness and applicability for test-time scaling. We also highlight that the observed improvement of $1e^{-2}$ on DrawBench is a substantively important result. **As demonstrated in previous work [2,3], improvements of this scale on DrawBench are considered significant and represent a clear advancement in the field.**
>
> - **Superiority on Challenging Benchmarks (GenEval & DPGBench)**: To further demonstrate that EvoSearch is not merely a "variant" of particle sampling but a superior optimization strategy designed for test-time scaling, we evaluated it on fine-grained benchmarks like GenEval and DPGBench (see Appendix B.3). Given $200\times$ NFEs at test-time, EvoSearch w/ SD2.1 can reach a `0.92` accuracy on GenEval, while particle sampling plateaus at a 0.86 accuracy. This substantial margin confirms that EvoSearch effectively utilizes its evolutionary selection mechanism to expand the exploration space and preserve diversity (evidenced by Table 2), solving complex prompts that standard particle sampling fails to handle.
> - **Qualitative Validation**: These quantitative gains are corroborated by the qualitative comparisons in Figs. 15-17 (Appendix B.4.1) and the video demos available on our project website (evosearch.github.io). These visualizations clearly highlight EvoSearch’s ability to generate more coherent and prompt-aligned images compared to the baseline.
>
> **Reference**
>
> [1] Singhal, Raghav, et al. "A general framework for inference-time scaling and steering of diffusion models.", ICML 2025
>
> [2] Ma et.al., Inference-Time Scaling for Diffusion Models beyond Scaling Denoising Step, CVPR 2025
>
> [3] Zhou et.al., Golden Noise for Diffusion Models: A Learning Framework, ICCV 2025.

---

> > ### Author Response · Authors · 2025-11-20
> > **Response to Reviewer XkkV (Part 2)**
> >
> > >2. ...raising concerns about practicality...
> >
> > Thanks for your feedback on the computational demands of EvoSearch.
> >
> > We would like to clarify that our paper focuses on **advancing the field of test-time scaling in visual generation** (including both image and video). Similar to how training-time scaling laws utilize massive compute to improve base models, test-time scaling investigates how much generation performance can be improved by trading off computation for quality during inference. Therefore, **requiring additional computational budget is an inherent characteristic of this research direction, not a limitation**. Our goal is to determine the upper bounds of generation quality when compute is not the bottleneck (e.g., for commercial video production or high-end asset production), rather than merely developing faster inference methods.
> >
> > While we explore the boundaries at high NFEs, **EvoSearch is also effective at lower budgets**. We respectfully point out that EvoSearch does not require 5000 NFEs to surpass baselines: As shown in Table 5 in Appendix B.3, given only $10\times$ NFEs ($\approx$ 1 minute for generation), EvoSearch (w/ Flux.1-dev) reaches a `93.51` score on DPGBench, significantly outperforming the particle sampling baseline (89.32). Please note that DPGBench contains 1065 prompts that comprehensively evaluate the text-to-image generative models from diverse dimensions.
> >
> > Additionally, as shown in Fig. 5, EvoSearch can outperform baselines given 500 NFEs, and the advantages of EvoSearch become more obvious as the NFEs increase (as claimed by the reviewer). We remark that this scalability is a critical property for test-time scaling methods. While baseline methods plateau, EvoSearch continues to convert additional compute into quality improvements.
> >
> > **Flexibility and Broad Applicability**
> >
> > EvoSearch provides the first unified framework for both flow-based and diffusion-based models across image and video tasks, eliminating the need for model-specific manual tuning. Furthermore, **EvoSearch is practically flexible**; users can adjust the evolution schedule and population size according to their specific test-time computation constraints. For example, users can reduce the number of evolution generations or the population size at each generation for a trade-off between quality and computation cost.
> >
> > **Enables a small-scale model to surpass a larger-scale model**
> >
> > We demonstrate a critical practicality in our experiments: EvoSearch allows a small model (Wan-1.3B) to outperform a much larger model (Wan-14B), as shown in lines 487-497 in our paper. Running EvoSearch on a 1.3B model is often computationally cheaper (and has lower VRAM requirements) than running a single inference pass on a 14B model, yet it achieves superior results. This proves that EvoSearch offers a practical path to state-of-the-art visual generation quality without requiring massive proprietary models.

---

> > > ### Author Response · Authors · 2025-11-20
> > > **Response to Reviewer XkkV (Part 3)**
> > >
> > > >3. ...The fact that EvoSearch does not compare against any such differentiable methods is a notable omission. It’s unclear if the authors avoided gradients due to concerns like non-convexity or local optima...
> > >
> > > We thank the reviewer for raising this insightful point regarding gradient-based alternatives. We deliberately chose a gradient-free evolutionary strategy for three key reasons:
> > > - **Universality**: Not all reward functions are differentiable (e.g., rewards from proprietary APIs, discrete metrics, or human feedback). A gradient-free approach makes EvoSearch universally applicable.
> > > - **Memory & Efficiency**: Backpropagating gradients through the diffusion process is memory-intensive and computationally expensive. Training-free methods like test-time scaling are significantly cheaper, allowing us to explore a wider search space within the same wall-clock time.
> > > - **Avoiding Local Optima**: Gradient ascent on noise latent space is prone to getting stuck in local optima or generating "adversarial" examples (e.g., high reward score but poor visual quality), which is a well-known problem in the literature. Our proposed EvoSearch, which is based on a gradient-free evolutionary algorithm, is better suited for non-convex landscapes as it maintains population diversity explicitly.
> > >
> > > To validate these claims, we followed the reviewer's suggestion and compared EvoSearch against ReNO (Eyring et al., 2024), a representative gradient-based method. We follow the official ReNO implementation. Both ReNO and EvoSearch employ the combination of the reward models (HPS, ImageReward, CLIP, PickScore) as the guidance, with SDXL-Turbo as the base model. We compare based on Wall-Clock Time to ensure a fair assessment of practical utility. ReNO requires ~33s for 50 iterations on a single GPU.
> > > | Methods | Computation / Time|GenEval|
> > > | - | -|-|
> > > | SDXL-Turbo (Base)| -|0.54|
> > > |+ReNO |50 iters / ~33s|0.65|
> > > |+EVOSearch |190 NFEs / ~33s|**0.71**|
> > > |+ReNO |500 iters / ~330s|0.68|
> > > |+EVOSearch |1000 NFEs / ~172s|**0.77**|
> > >
> > > From the results shown in the Table, we have the following observations：
> > > - **Superior Efficiency**: Under the same wall-clock time constraint (~33s), EvoSearch significantly outperforms ReNO (0.71 vs. 0.65). Because EvoSearch is gradient-free, we can perform nearly $4\times$ as many evaluations (190 NFEs) in the time ReNO takes for 50 gradient steps, allowing for more extensive exploration.
> > > - **Better Scalability**: ReNO suffers from diminishing returns, improving only slightly (0.65 $\to$ 0.68) even when computation is increased by $10\times$. In contrast, EvoSearch effectively converts increased compute into quality, reaching a `0.77` score.
> > > - **Robustness**: The plateauing of ReNO suggests it likely falls into local optima or over-optimizes the reward model without improving visual fidelity (the "adversarial noise" problem). EvoSearch avoids this by maintaining a diverse population of candidates, ensuring that the optimization yields actual perceptual improvements rather than just exploiting the reward model gradients.

---

> > > > ### Author Response · Authors · 2025-11-20
> > > > **Response to Reviewer XkkV (Part 4)**
> > > >
> > > > >4. ...The authors did not include a comparison with the latest particle sampling advancement, namely the Rollover Budget Forcing (RBF) ...
> > > >
> > > > Thanks for your good suggestion to improve our work! We would like to clarify that Rollover Budget Forcing (RBF) is primarily designed for flow-based models in image generation. In contrast, EvoSearch is a unified framework that generalizes effectively across both diffusion and flow models, as well as image and video generation tasks. This universality is a key contribution of our work.
> > > >
> > > > To further demonstrate the advantages of EvoSearch, we have added RBF for comparison. Following RBF's official implementation, we choose Flux as the base model with ImageReward as the guidance. We fix the total number of function evaluations (NFEs) to 500 and set the number of denoising steps to 10. We evaluate both EvoSearch and RBF on DrawBench. While the numerical margins may appear modest, we emphasize that on the DrawBench scale, a consistent improvement across 200 prompts on three different metrics (including both target and unseen metrics) confirms that EvoSearch generates strictly superior samples.
> > > >
> > > > | Methods | ImageReward (target) | ClipScore (o.o.d.) |HPSv2(o.o.d.) |
> > > > | - | - |-|-|
> > > > | RBF    | 1.38  |0.281|0.305|
> > > > | EvoSearch    | **1.41**  |**0.284**|**0.310**|
> > > >
> > > > The performance gap can be attributed to the fundamental difference in design philosophy:
> > > > - **RBF (Passive Filtering)**: RBF operates as a "passive" selection method. It optimizes the budget by terminating low-quality paths early and reallocating compute to promising ones. However, it is limited by the quality of the initial samples; it cannot create new, better trajectories actively along the denoising process.
> > > > - **EvoSearch (Active Exploration)**: EvoSearch treats the denoising trajectory as an evolutionary path. Through our mutation mechanisms, EvoSearch actively explores the latent space, creating new candidate states beyond the pre-trained model's distribution. This allows EvoSearch to discover higher-quality solutions that RBF would simply miss.
> > > >
> > > > >5. ...The paper aspires to present a general-purpose test-time scaling approach, but in doing so it may be spreading its contributions thin. ...
> > > >
> > > > We thank the reviewer for recognizing the novelty and compelling nature of our results in the video generation domain! We agree that the video results are a highlight of this work. However, we respectfully disagree that the image generation contribution is weak. We validate the effectiveness of EvoSearch against a wider array of established baselines (like FK-Steering, ReNO, and RBF). As detailed in our previous responses, EvoSearch achieves SOTA results on image benchmarks (DrawBench, GenEval, DPGBench), consistently outperforming strong baselines (FK-Steering) and newly-added methods (RBF, ReNO). We believe the `universality` of EvoSearch is exactly what makes it a strong contribution.
> > > >
> > > > As suggested, we will revise the manuscript to place greater emphasis on the video domain. Our results here are indeed significant: (1) We utilize widely adopted metrics from VBench and VBench2 to demonstrate that EvoSearch generalizes well to out-of-distribution assessments, ensuring robust performance beyond the optimization target. (2) A critical finding is that EvoSearch enables a small model (Wan-1.3B) to surpass the performance of a model an order of magnitude larger (Wan-14B) on key metrics. `This perfectly validates the core premise of test-time scaling: trading inference compute for model parameter efficiency.`
> > > >
> > > > >6. Have you considered leveraging gradient-based optimization on the reward function instead of (or in addition to) evolutionary search?
> > > >
> > > > We have compared with the gradient-based optimization approach, like ReNO. We also updated our paper to include the comparison and discussions about the gradient-based methods in Appendix D.

---

> > > > > ### Author Response · Authors · 2025-11-20
> > > > > **Response to Reviewer XkkV (Part 5)**
> > > > >
> > > > > >7. The results indicate EvoSearch can require thousands of steps (and very long wall-clock time) to fully exploit the reward. How practical is this method, and are there ways to reduce the compute required?
> > > > >
> > > > > We thank the reviewer for this practical question. We argue that EvoSearch is not only scalable but also highly efficient compared to baselines when measuring the "time-to-result."
> > > > >
> > > > > As established in our previous responses, EvoSearch is highly adaptable. While it scales effectively to thousands of steps, it also yields State-of-the-Art results with limited budgets (e.g., surpassing baselines with only $10\times$ NFEs, as shown in Table 5). This flexibility allows users to tune the method based on their specific latency constraints.
> > > > >
> > > > > To further demonstrate the computation efficiency of our method, we added additional experiments to compare the required wall-clock time and NFEs of different methods to reach the same ImageReward (i.e., 1.46) on DrawBench. We record the wall-clock time on the same machine to ensure fairness. **The following experiments demonstrate that EvoSearch costs less time than all the baselines, showing the superior computation efficiency of our method**. Even compared to the recent RBF method, EvoSearch is more efficient. RBF consumed more time (92.1s) yet achieved a slightly lower score (1.45), further validating EvoSearch's practical advantage. The efficiency of EvoSearch stems from its streamlined fitness evaluation mechanism: EvoSearch requires only one reward computation per sample (at the end of the denoising generation). In contrast, RBF and particle sampling methods demand multiple reward computations and resampling operations per sample at multiple intermediate steps, resulting in significantly higher computational overhead.
> > > > > | Methods|Required NFEs|Required Wall-Clock Time |ImageReward on DrawBench|
> > > > > | -| -| -|-|
> > > > > | EvoSearch|	$50\times 90$|	**85.6s**   |1.46|
> > > > > | RBF|	$50\times 90$|	92.1s    |1.45|
> > > > > | Particle Sampling|	$50\times 100$|	117.7s    |1.46|
> > > > > | Best of N|	$50\times 1200$|	971.4s    |1.46|
> > > > >
> > > > >
> > > > > To reduce the test-time computation, users can adjust the evolution schedule $\mathcal{T}$ and population size $K$ according to their specific test-time computation constraints. For example, users can reduce the number of evolution generations or the population size at each generation for a trade-off between quality and computation cost.
> > > > >
> > > > > ---
> > > > >
> > > > > **Overall, we hope that our responses above have addressed your concerns. If they have, we would appreciate it if you could kindly consider raising the score. If not, we are always willing to address any of your further concerns. Thank you again for reviewing our work!**

---

> ### Author Response · Authors · 2025-11-26
> **Looking forward to your reply!**
>
> Dear Reviewer XkkV,
>
> Considering there have been more than 5 days since we posted our response, we'd like to ask if we have addressed your concerns. We believe that we have clarified the concerns and questions raised.
>
> Specifically, we addressed your comments by clarifying our advantages over previous methods in detail, the practicality of test-time scaling methods, adding new experiments to compare with the gradient-based ReNO method (`see Appendix D`), adding new experiments to compare with RBF (`see Appendix E`), and adding new experiments to further validate the computational efficiency of our method.
>
> This will be a good improvement to our manuscript. We are happy to provide further clarification if you have any additional concerns. Thanks again for your feedback.
>
> Best,
>
> The authors

---

### Author Response · Authors · 2025-11-20
**General Response**

We thank all of the reviewers for their time and insightful comments. Furthermore, we are very glad to find that reviewers generally recognize the effectiveness, novelty, and significance of our work:
- **Method**: EvoSearch introduces a fresh perspective by framing inference-time optimization as an evolutionary process [**XkkV,KnUy**], which is an original way to maintain diversity and avoid the collapse seen in earlier methods [**XkkV,oKyW**]. It is among the first to study test-time scaling for video models [**XkkV,6YMr**].
- **Experiments**: The experiments show notable gains in output metrics [**XkkV,oKyW,KnUy,6YMr**]. It has general applicability [**6YMr**] and no training overhead [**oKyW**]. The paper provides extensive analysis including toy example and ablation study with hyper-parameters [**KnUy**].

Meanwhile, we thank all the reviewers for their helpful and constructive feedback to improve the quality of our work again. We have carefully updated our paper to incorporate the valuable suggestions from the reviewers. We summarize the revisions and added experiments as follows:
- [**XkkV**] We add comparison and discussions with gradient-based methods like ReNO in Appendix D.
- [**XkkV**] We add comparison and discussions with ROLLOVER BUDGET FORCING (RBF) in Appendix E.
- [**XkkV,KnUy,6YMr**] We add additional experiments on computational efficiency.
- [**oKyW**] We made several updates on the figure, caption, and pseudocode to improve clarity.
- [**KnUy**] We updated the related work section for more discussions.

---

All changes in the new PDF are highlighted in red.

We hope to have addressed all the raised concerns and would be happy to respond to further questions and suggestions.

---

### Author Response · Authors · 2025-12-03
**Letter to AC, SAC, and PC: Summarization of the discussion phase (Part 1)**

Dear AC, SAC, and PC,

We sincerely thank you for your time and effort in handling our paper! We would like to summarize the key points from the reviews, our rebuttal, the feedback received, and the consensus status (after our rebuttal) for your consideration.

## Summary of reviewers' recognitions
The reviewers reach a positive consensus on the primary strengths of our work. The core contribution of EvoSearch, its **novel** framing of inference-time optimization as an evolutionary process, is widely recognized (**XkkV, KnUy, oKyW**). Furthermore, reviewers acknowledge the **broad impact** of our approach across **both diffusion and flow-based models**, and its position as **the first work** to **explore test-time scaling** for **video models** (**XkkV, oKyW, 6YMr**). All reviewers recognize the significant performance improvements of our method. They specifically commended its general applicability (**6YMr**) and its practical advantage of having no training overhead (**oKyW**). The quality of our experimental validation, including the toy examples and hyperparameter ablations, was also highlighted as a positive aspect (**KnUy**).

## Summary of reviewers' concerns and our additional results and responses

Two reviewers (**oKyW, KnUy**) gave positive ratings, recognizing both the novelty and the impressive performance of our work. Regarding the negative ratings, we believe the concerns raised by Reviewers **XkkV** and **6YMr** stem from specific misunderstandings that we have clarified in our responses. Specifically, **Reviewer 6YMr incorrectly characterized our work as relying on heuristic choices and misunderstood the main techniques of our method**; we have clarified the core innovations and rationale behind our method to correct this. Although there was no active engagement from reviewers during the discussion phase, we firmly believe we have fully addressed their concerns in our rebuttal and the revised manuscript (changes highlighted in red). Below, we summarize the main concerns raised and how our rebuttal addresses them:

>  **Reviewer XkkV**: The practical advantages over existing particle sampling methods.

We provided detailed empirical results to validate the **superior efficacy** of our method compared with previous SOTA methods, including distinct empirical performance on different benchmarks (e.g., DrawBench, GenEval, DPGBench), different metrics (e.g., ImageReward, ClipScore), different tasks (e.g., image generation and video generation), different base models (e.g., SD2.1, Flux, Wan, Hunyuan Video), and extensive qualitative demos（provided in Appendix B.4 and our anonymous project website).
> **Reviewer XkkV**: The practicality of test-time scaling methods.

We clarified the practicality and applicability of our proposed test-time scaling method, such as commercial video production or high-end asset production, where compute is not the bottleneck. We note that test-time scaling is an emerging research direction that receives increased attention from the vision community.

> **Reviewer XkkV**: Comparison with gradient-based methods and a relevant baseline.

We added comparison and discussions with gradient-based methods like ReNO [1] in Appendix D. As required by the reviewer, we compare with ROLLOVER BUDGET FORCING (RBF) [2] in Appendix E. Our method remains superior in scalability and provides a more significant improvement.

>**Reviewer oKyW**: The clarity of our presentation.

We carefully updated the figure, caption, and pseudocode to improve clarity.
>**Reviewer KnUy**: More discussions about related works.

As suggested by the reviewer, we updated the related work section to further demonstrate our novelty and distinct contributions compared with previous methods.


> **Reviewers KnUy, 6YMr**: The compuational efficiency of our method.

We added additional experiments on wall-clock time to demonstrate the superior computational efficiency and real-world usage of our method (see Appendix B.2).
> **Reviewer 6YMr**: Concerns regarding the heuristic choices of our method.

We clarified that our method is robust and scalable across different settings and requires negligible heuristic choices. We provided extensive evidence (as detailed in our rebuttal) to validate this point.
> **Reviewer 6YMr**: Concerns regarding the soundness of our method.

We have articulated the core design principles of our method and provided a detailed analysis of the mechanisms responsible for the superior empirical performance of EvoSearch.

---

> ### Author Response · Authors · 2025-12-03
> **Letter to AC, SAC, and PC: Summarization of the discussion phase (Part 2)**
>
> ## Summary of the contributions
> To help AC better evaluate our paper, **we highlight our contributions as follows**:
> - To our knowledge, we are the **first** to propose a unified test-time scaling framework for both diffusion and flow generative models across both image and video generation.
> - Inspired by evolutionary algorithms, we design specialized selection and mutation mechanisms tailored to the denoising process, effectively enhancing exploration while maintaining diversity. This is the key advantage and novelty of our method compared with previous methods.
> - We validate the superior performance of our method on `4` different model families (SD2.1, Flux, Wan2.1, Hunyuan) and `2` modalities (Image, Video) using `8` different reward functions (ImageReward, ClipScore, Aesthetic, HPSv2, GenEval, DPGBench, VBench, VBench2). Therefore, we provide a reproducible and widly-applicable framework for the research community.
> - Surprisingly, We find that our method enables small-scale models to be comparable to larger-scale models, allowing the Wan 1.3B model to achieve competitive performance with the 10× larger Wan 14B model within the same inference time (see Table 3). Therefore, our method offers great `potential` and `real-world practicability`.
>
> We greatly appreciate your careful consideration of our responses and the overall review situation in your final evaluation.
>
> Best,
>
> The authors
>
> **Reference**
>
> [1] Eyring, Luca, et al. "Reno: Enhancing one-step text-to-image models through reward-based noise optimization." NeurIPS 2024.
>
> [2] Kim, Jaihoon, et al. "Inference-time scaling for flow models via stochastic generation and rollover budget forcing." NeurIPS 2025.

---

### Meta-Review · Area_Chair_JFAK · 2025-12-19

**Summary:**

The major concerns that led to the final decision are the small improvements over the compared baselines and high computational cost, lack of comparisons with gradient-based methods and recent state-of-the-art method, and insufficient justification for the design. Some other concerns are mostly about presentation, clarity, and missing references.

**Reviewer Concerns:**

The concerns about presentation, clarity, and missing references were addressed by the rebuttal, and the author provided additional justification for the choices of the proposed design. However, while the rebuttal includes new experimental comparisons with ReNO and RBF, these results are relatively limited in scale and scope, and do not fully resolve the reviewers’ concerns about the method’s comparative advantages. Additional experiments are also needed to provide a comprehensive overhead/runtime comparison across methods. It would be better for the authors to take additional time to more thoroughly incorporate the suggested baselines and conduct more comprehensive experiments in a revised version of the paper.

**Reviewer Scores:**

Reviewer oKyW and KnUy are likely to keep the positive scores as their concerns were mostly about presentation and clarity. Reivewer XkkV and 6YMr are not likely to increase the score as the concerns were not fully addressed.

---

### Decision · Program_Chairs · 2026-01-26

Reject